# E³FORMER: AN ADAPTIVE ENERGY-AWARE ELASTIC EQUIVARIANT TRANSFORMER MODEL FOR PROTEIN REPRESENTATION LEARNING

## ABSTRACT

Structure-informed protein representation learning is essential for effective protein function annotation and *de novo* design. However, the presence of inherent noise in both crystal and AlphaFold-predicted structures poses significant challenges for existing methods in learning robust protein representations. To address these issues, we propose a novel equivariant Transformer-State Space Model(SSM) hybrid framework, termed $E^3$former, designed for efficient protein representation. Our approach leverages energy function-based receptive fields to construct proximity graphs and incorporates an equivariant high-tensor-elastic selective SSM within the transformer architecture. These components enable the model to adapt to complex atom interactions and extract geometric features with higher signal-to-noise ratios. Empirical results demonstrate that our model outperforms existing methods in structure-intensive tasks, such as inverse folding and binding site prediction, particularly when using predicted structures, owing to its enhanced tolerance to data deviation and noise. Our approach offers a novel perspective for conducting biological function research and drug discovery using noisy protein structure data. Our code is available on https://anonymous.4open.science/r/E3former-207E.

## 1 INTRODUCTION

Protein representation learning plays a crucial role in advancing our understanding and application of the structural and biological functions of proteins. A wide array of protein-related tasks, such as predicting interactions, annotating functions, and designing protein binders, depend heavily on the development of robust protein representations. (Tubiana et al., 2022) (Bushuiev et al., 2023) (Lisanza et al., 2023) (Gligorijević et al., 2021).

The recent advancements in experimental technologies, coupled with the groundbreaking development of protein structure prediction models like AlphaFold (Jumper et al., 2021), have significantly increased the availability of detailed protein structural data. This surge in data has shifted the focus of protein representation learning towards effectively harnessing this rich structural information. However, the existence of inherent noise in crystal and AlphaFold-predicted structures (Acharya & Lloyd, 2005; Moore et al., 2022) presents substantial challenges for current methods

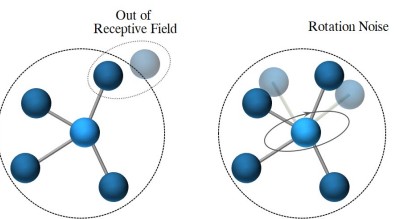

Figure 1: Noise mechanism in equivariant GNN based on cut-off radius graph.

in learning robust protein representations. In response to these challenges, the development of geometric deep learning approaches that are tolerant to noise or the deviations in protein structures data has become crucial.

Since the latest progress in equivariant neural networks has demonstrated their good ability in handling diverse protein structure data (Duval et al., 2023), exploring the robustness representation learning of structural information based on this framework has emerged as a pivotal breakthrough. Although existing equivariant deep learning models like EGNN (Satorras et al., 2021), GearNet (Zhang et al., 2022), spherical harmonics-based models (Liao & Smidt, 2022) and protein specific model GCPNet (Morehead & Cheng, 2024) architectures exhibit strong performance, they still show high sensitive to data bias or noise (Joshi et al., 2023). This paper aims to develop a model tailored for representing protein macromolecular data within the equivariant neural network framework. The model can adaptively alleviate the effects of noise and data bias, while also tackling the dynamic of macromolecules.

The primary factor contributing to model sensitivity to noise is the offset of atomic positions in 3D Euclidean space. On the one hand, the model's estimation of node attributes is significantly

influenced by its neighboring nodes chosen, and perturbation in node coordinates may affect the model's selection of these. On the other hand, unlike invariant methods, models incorporating prior knowledge of geometric equivariance are more susceptible to rotation (Li et al., 2024). Further explanation on this topic is provided in Figure 1.

Herein, we introduce $E^3$former, an Equivariant Transformer-SSM hybrid architecture that incorporates an Energy-aware radius graph and an Elastic selective mechanism. And we applied our model to two types of datasets with different structural sources: one predicted by AlphaFold2 and the other derived from experimental crystal structures, in order to assess whether our model can robustly tolerate noise in protein structure data. These datasets encompass two structure-sensitive tasks—Protein-Protein Interaction(PPI) tasks (Gainza et al., 2020) and inverse folding tasks (Ingraham et al., 2019) to evaluate the model's effectiveness in tasks that rely on geometric features.

Our contribution can be summarized as follows: (1) Inspired by molecular dynamics (Geada et al., 2018), $E^3$former leverages an **energy-aware radius function** and **radius sampler** to adaptively modify the receptive field based on the atoms environment in 3D Euclidean space, thereby mitigating the effects of data biases on constructing protein proximity graphs. (2) **A novel equivariant elastic selective SSM** is proposed to extract and compress high-order tensors that are particularly sensitive to rotations. We use the parameter-sharing SSM module as the sparse representation of Transformer (Behrouz & Hashemi, 2024), and utilize spherical harmonics-based models to handle irreducible representations in tensors of various orders. By performing a separate sparse representation of the high-dimensional tensor and processing it with a parameter-sharing matrix, the model can enhance the information encoding of biased data with a heightened signal-to-noise ratio. (3) **Creating a new dataset version based on the Alphafold structure** for established public tasks to systematically assess the tolerance of each model to data bias and noise. (4) Empirical results demonstrate that our model achieves overall better results across all tasks compared to previous state-of-the-art methods on the Alphafold-predicted datasets (Table 2). Benefiting from its anti-noise capabilities, the model outperformed the state-of-the-art(SOTA) methods by 11.2% in the inverse folding task. In the experimental data, $E^3$former continues to maintain SOTA performance in various tasks owing to its enhanced information extraction capabilities. These outcomes showcase that our approach learns robust protein representations that can address biases in prediction data and inherent noise present in crystal structures.

## 2 RELATED WORK

### 2.1 STRUCTURED-BASED PROTEIN REPRESENTATION LEARNING

With advancements in experimental and structure prediction technologies, the availability of protein structure data has significantly expanded (Jumper et al., 2021) (wwPDB consortium, 2018) (Baek et al., 2021). A range of representation learning approaches for protein structures have emerged. **Voxelized representation-based** methods map the three-dimensional structure of proteins into voxelized 3D volumes and encode the atomic system using techniques such as 3D convolutional neural networks(3DCNNS). For instance, a series of methods like (Pagès et al., 2019) (Anand et al., 2022) (Liu et al., 2021)leverage 3DCNNs to encode protein structures, demonstrating the efficacy of this representation and encoding approach across diverse tasks. Furthermore, Metal3D (Dürr et al., 2023) integrates multiple physical and chemical properties as inputs based on this representation, enriching the environmental information within the framework of 3DCNNs. On the other line, **Graph-structured representations-based** methods for protein structure involve mapping the structure of proteins as a proximity graph over amino acid nodes, leveraging Graph Neural Networks (GNNs) to capture intricate interactions among nodes (Han et al., 2024). GVP Jing et al. (2020) employing equivariant GNNs for computational protein design and model quality assessment. Methods like Schnet (Schütt et al., 2018) and Scannet (Tubiana et al., 2022)integrate 3D spatial information and chemical features of atoms within GNN frameworks, applying these models to tasks such as protein binding site prediction. EGNN (Satorras et al., 2021), GearNet-Edge (Zhang et al., 2022), GCPNet (Morehead & Cheng, 2024), and other methods have also made significant architectural advancements in GNNs, enhancing their capabilities in modeling complex relationships within protein structures. Although powerful, these models are mainly designed for rigid experimental structures and struggle with the inherent noise present in real-world protein applications.

### 2.2 EQUIVARIANT NEURAL NETWORKS

Equivariant neural networks have recently demonstrated remarkable success in modeling 3D atomic systems, encompassing chemical small molecules and biological macromolecules (Fuchs et al., 2020)

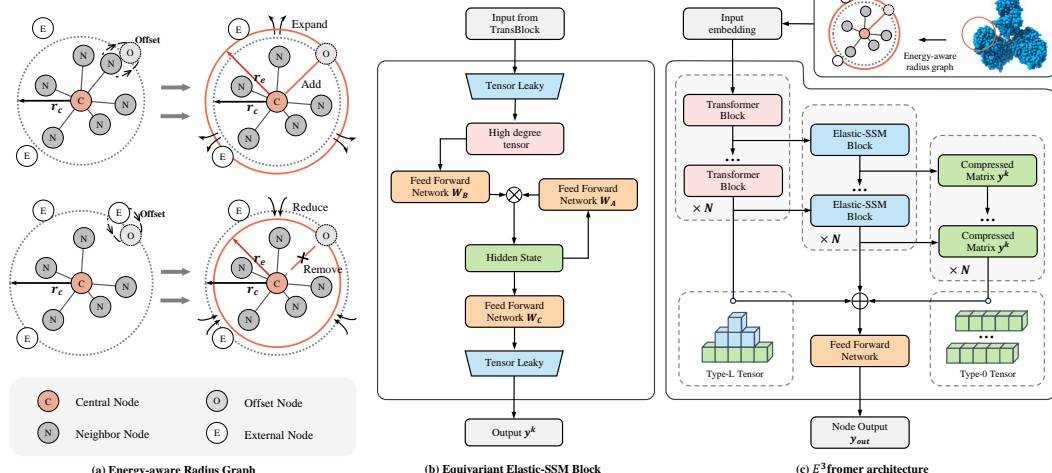

Figure 2: Overview of $E^3$former. (a) Energy-Aware Radius Graph, for nodes unexpectedly leave or enter the receptive field due to structural noise, model adjusts the radius to correct the adjacency relationships. (b) Equivariant Elastic-SSM Block, high tensor-leaky module is designed to filter high-order tensors. (c) $E^3$former architecture, consisting of an energy-based receptive radius graph and a hybrid Transformer-SSM module, designed to learn representations of protein structural information.

(Batzner et al., 2022) (Liao & Smidt, 2022) (Liao et al., 2023) (Townshend et al., 2021) (Bushuiev et al., 2023). Among these, Cartesian-based equivariant neural networks focus on modeling 3D molecular graphs in Cartesian coordinates (Xu et al., 2021). This approach involves updating and exchanging messages between scalars and vectors concurrently, transforming vectors into Cartesian tensors, and confining operations within these tensors to maintain equivariance. For example, the GVP method (Jing et al., 2020) segregates atomic features within protein data into scalars and vectors, executing equivariant message passing. TorchMD-Net (Thölke & De Fabritiis, 2022), an equivariant transformer-based Graph Neural Network (GNN), employs an attention mechanism for weighted message propagation. TensorNet model (Simeon & De Fabritiis, 2024) Cartesian tensors to higher ranks, enhancing the expressive power of the model for equivariant message passing tasks.

Spherical harmonics-based models, on the other hand, leverage spherical harmonics functions and irreducible representations to flexibly process data while maintaining equivariance. These models decompose spherical tensors into different degrees, showcasing robust fitting capabilities in 3D molecular datasets (Thomas et al., 2018) (Brandstetter et al., 2021).The classical TFN (Thomas et al., 2018) method harnesses filters constructed from spherical harmonics, enabling the conversion of data into versatile higher-order tensors. A SE(3)-Transformers method is proposed (Fuchs et al., 2020) to adapt the TFN framework to the self-attention operation during aggregation. Equiformer (Liao & Smidt, 2022) and Equiformerv2 (Liao et al., 2023) integrates equivariant graph attention neural networks into Transformer-style blocks.

While the aforementioned equivariant methods enhance the model's applicability to 3D atom systems, some struggle to capture more intricate interactions due to the lack of utilization of high-order tensors, while others exhibit high sensitivity to noise when employing these high-order tensors.

## 3 $E^3$FORMER

In this section, we introduce an adaptive **E**nergy-aware **E**lastic **E**quivariant Transformer-SSM hybrid architecture, termed $E^3$former. The energy-aware radius graph module will be detailed in Section 3.1 , while the high-order tensor elastic compression SSM module and the discussion of equivariance will be described in Section 3.2. The overall architecture and designed equivariant operations will be discussed in Section 3.3. The $E^3$former method, incorporating these two innovative modules, is illustrated in Figure 2.

### 3.1 ENERGY-AWARE PROTEIN GRAPH

We represent the 3D protein structures as a connected graph at the residue-level, denoted as graph $G = (V, E)$, where nodes $V$ represent the amino acids within the protein graph, and edges $E$ signify the interactions among them. Typically, protein graphs employ a local radius cut-off with $k$-nearest neighbors. Given a preset distance cutoff $r^c$, the edge set is:

$$E = \{e_{j \to i}\}_{i \neq j, j \in \mathcal{N}_c(i)}, \mathcal{N}_c(i) = d_{ij} \leq r^c \text{ and } j \in \mathcal{N}_{\text{top-}k}(i), \tag{1}$$

where $d_{ij}$ denoting the distance between nodes $i$ and $j$, $\mathcal{N}_{\text{top-}k}(i)$ represents the top $K$ nodes closest to node $i$, $\mathcal{N}_c(i)$ means the chosen neighbors of node $i$ In this context, the strong inductive bias of locality opts for a restricted and typically more relevant neighbor for the node. However, within the structure predicted by AlphaFold, the data noise and protein flexibility can induce perturbation in node coordinates, potentially leading to the neglect of crucial edges in the local radius cut-off graph, as illustrated in Figure 2(a).

**Energy-aware radius function** To solve the above problems, we proposed an energy-aware radius graph module for adaptively adjusting the receptive field of protein graphs based on energy. This allows the model to expand the receptive field when important neighbor nodes are far away due to noise or flexibility, and shrink the receptive field when their neighbor nodes are too dense, thereby helping the model to model locality more reasonably under noisy conditions. We used the commonly used Lennard-Jones potential function to describe the main interactions between molecules, which are:

$$\mathcal{E}_i = \epsilon_c \left[ \left( \frac{\sigma}{d_{ij}} \right)^{12} - \left( \frac{\sigma}{d_{ij}} \right)^{6} \right], \tag{2}$$

where $e_i$ is the sum of the L-J potential energy of node $i$. In order to reflect the model's insensitivity to this parameter, we set the potential energy parameters $\epsilon$ and $\sigma$ to fixed constants in all tasks. $d_{ij}$ represents the distance between nodes. After calculating the sum of the potential energy $e_i$ of node i, its adaptive radius is positively correlated with $e_i$, as follows:

$$r_i^{field} \propto \mathcal{E}_i. \tag{3}$$

**Energy-based radius sampler.** If we were to assign a static radius to individual nodes, the characteristic of "the presence of a distant neighbor" would be conveyed to the node via the radial basis function in the model, imparting an excessively robust inductive bias. To mitigate this issue, we have introduced an energy-driven radius sampler. Initially, we calculate the normalized potential energy of each node $\mathcal{E}^{norm}$:

$$\mathcal{E}_i^{norm} = \frac{\log \left( \mathcal{E}_i - \mathcal{E}_{min} + 1 \right)}{\log \left( \mathcal{E}_{max} - \mathcal{E}_i + 1 \right)}, \tag{4}$$

among them, $\mathcal{E}_{min}$, $\mathcal{E}_{max}$ is the minimum or maximum value of the node potential energy in the entire protein, and the logarithmic operation is used to smooth the radius distribution of different nodes.

Based on the normalized potential energy $\mathcal{E}^{norm}$, the sampling function is:

$$r_i^{\mathcal{E}} \in Beta(\alpha + \beta \mathcal{E}_i^{norm}, \alpha - \beta \mathcal{E}_i^{norm}). \tag{5}$$

To constrain the model's sampling radius within a predetermined range, we leverage the $Beta$ distribution, ensuring that nodes with lower normalized potential energy are more likely to possess a larger receptive field. Here, $\alpha$ and $\beta$ are constants, both set to identical values across all tasks to demonstrate the model's resilience to this parameter.

---

**Algorithm 1** Energy-based radius sampling process

> **input:** coordinate set $X = \{x_1, x_2, ..., x_N\}$, cut-off radius $r_c$, max neighbors number $k$
> Initialize energy set $\mathcal{E} \leftarrow \emptyset$
> Initialize energy radius set $\mathcal{R} \leftarrow \emptyset$
> **for** $i = 1$ **to** $N$ **do**
>   $\mathcal{N}_c(i) = d_{ij} \leq r_c$ and $j \in \mathcal{N}_{\text{top-}k}(i)$.
>   $\mathcal{E}_i = \epsilon_c \left[ \left( \frac{\sigma}{d_{ij}} \right)^{12} - \left( \frac{\sigma}{d_{ij}} \right)^{6} \right]$ and $j \in \mathcal{N}_c(i)$.
>   $\mathcal{E} \leftarrow \mathcal{E} \cup \{\mathcal{E}_i\}$.
> **end for**
> **for** $i = 1$ **to** $N$ **do**
>   $\mathcal{E}_i^{norm} = \frac{\log(\mathcal{E}_i - \mathcal{E}_{min} + 1)}{\log(\mathcal{E}_{max} - \mathcal{E}_i + 1)}$.
>   $r_i^{\mathcal{E}} \in Beta(\alpha + \beta \mathcal{E}_i^{norm}, \alpha - \beta \mathcal{E}_i^{norm})$.
>   $\mathcal{R} \leftarrow \mathcal{R} \cup \{r_i^{\mathcal{E}}\}$.
> **end for**
> **return** $\mathcal{R}$

---

## 3.2 EQUIVARIANT HIGH-TENSOR-ELASTIC SELECTIVE SSM

In equivariant method based on irreducible representation, the sensitivity to geometric coordinate rotation increases with the degree $L$ of the tensor (Liao & Smidt, 2022). For type-$L$ vectors, $L = 0$ denotes a scalar while $L = 1$ signifies Euclidean vectors. To mitigate the amplification of geometric feature noise in the high-order channels of the equivariant transformer, drawing inspiration from the chosen space state model, we introduce an equivariant elastic high-tensor SSM block. This block, utilized in conjunction with the equivariant transformer module, aims to extract the high-order tensor in the irreducible representation and integrate it into the equivariant elastic-SSM module. This integration replaces the fully-connected equivariant Transformer attention with its sparse alternatives (Behrouz & Hashemi, 2024). The interplay between these two blocks is illustrated in Figure 2(c).

**High-tensor leaky layers.** In an N-layer equivariant transformer neural network, the output of each layer is denoted as $Tk^{out}$. These outputs collectively create a sequence $\{T_k^{out}|k = 1, 2, ..., N\}$ which subsequently serves as the input $xk^{in}$ for the Elastic SSM after traversing through a high-tensor leaky layer:

$$X_k^{in} = L_{(Lmin, Lmax)}^{leaky}(T_k^{out}). \tag{6}$$

$L_{(L_{min}, L_{max})}^{leaky}$ signifies that following the depth-wise tensor product operation with the learnable parameter layer, solely the tensors ranging from degree $L_{min}$ to $L_{max}$ are preserved.

**Equivariant elastic selective SSM.** SSM is a model that employs a linear Ordinary Differential Equation (ODE) to map the input sequence $x(t)$ combined with the hidden state vector $h(t)$ to the output $y(t)$ Its basic form is:

$$h'(t) = \boldsymbol{A}h(t) + \boldsymbol{B}x(t),$$
$$y(t) = \boldsymbol{C}h(t), \tag{7}$$

Where $\boldsymbol{A}$, $\boldsymbol{B}$, and $\boldsymbol{C}$ represent the state matrix, input matrix, and output matrix, respectively. In the elastic-SSM module, we substitute the input of the discretized time with the model input $\{X_k^{out}|k = 1, 2, ..., N\}$:

$$h_t = h_k = \bar{\boldsymbol{A}}h_{k-1} + \bar{\boldsymbol{B}}x_{k-1},$$
$$yt = y_k = \bar{\boldsymbol{C}}h_k, \tag{8}$$

Furthermore, we utilize the matrices $\boldsymbol{W}_A$, $\boldsymbol{W}_B$, $\boldsymbol{W}_C$ that share learnable parameters to replace $\boldsymbol{A}$,$\boldsymbol{B}$,$\boldsymbol{C}$, thereby introducing a selection mechanism. This allows the model to dynamically extract essential information from the tensor.

$$h_k = (\boldsymbol{W}_A \otimes_{dtp} h_{k-1}) \otimes_{dtp} (\boldsymbol{W}_B \otimes_{dtp} X_k^{out}),$$
$$y_k = L_{(0,0)}^{leaky}(\boldsymbol{W}_C \otimes_{dtp} h_{k-1}), \tag{9}$$

The Depth-wise tensor product $\otimes_{dtp}$ is utilized to define the number of output channels and keep the equivariance of the operation (Liao & Smidt, 2022). The $L_{(0,0)}^{leaky}$ operation is employed to condense the output of the SSM module into a scalar, enabling the extraction of stable signals from noise. The results of several elastic-SSM blocks will be concatenated and employed in conjunction with the output of the Transformer block as the ultimate output:

$$\boldsymbol{Y} = (y_1||y_2||...||y_k), \boldsymbol{Z} = (\boldsymbol{Y}||\boldsymbol{T}_k), \tag{10}$$

where $(\cdot\|\cdot)$ is the concatenation operation and equivariant elastic-high-tensor SSM block as illustrated in Figure 2(b)

***Proof of Equivariance.*** For function tensor leaky $f$ with input $x$, we need prove that: for $F : X \rightarrow Y$ mapping between tensor spaces $X$ and $Y$. Given a group $G$ and group representations $D_x(g)$ and $D_y(g)$, we have $D_Y(g)f(X) = f(D_X(g)x)$, for any $x \in X, y \in Y, g \in G$. And for $f(X)$:

$$f(x) = L_{(L_1, L_2)}^{leaky}(\boldsymbol{W}_C \otimes_{dtp} X) \tag{11}$$

according to the definition of flexible tensor product for irreps in **e3nn** library (Geiger & Smidt, 2022) and equivariance proven by tensor product, we have:

$$\boldsymbol{W}_C \otimes_{dtp} D_X(g)x = D_Y(g)y'. \tag{12}$$

For $L_{(L_1, L_2)}^{leaky}$ operation, it preserves the selected tensor channels from $L_1$ to $L_2$. Because addition and deletion operations on tensors from different channels are rotation- and translation-independent, they will not break the equivariance:

$$L_{(L_1, L_2)}^{leaky}(D_Y(g)y') = D_Y(g)y \tag{13}$$

Besides, by utilizing other operation proven to equivariance, $E3$former demonstrates invariance for node representations in scalar type and E(3)-equivariance for 3D coordinates in vector type. For further exploration of the properties of the $E_3$former, please refer to the Appendix B for a detailed proof.

## 3.3 ARCHITECTURE OF $E^3$FORMER

The $E^3$former architecture comprises an energy graph module (Section 3.1), an equivariant SSM (Section 3.2), and a $SE(3)$-equivariant transformer module(Equiformer) (Liao & Smidt, 2022). In the context of a protein graph $G = (V, E)$, each node $v_i \in V$ corresponds to an amino acid. To mitigate computational complexity, we adopt a $C_\alpha$-based graph representation method as the foundation and integrate various featurization strategies to incorporate additional structural insights (Jamasb et al., 2024). The node $v_i$ possesses a feature encoding $h_v^i$ and the following features include:

- Residue Type (not used in inverse folding task), a 16-dimensional transformer-like positional Encoding (Vaswani, 2017).

- Backbone dihedral angles $\phi, \psi, \omega \in \mathbb{R}^6$.

- Virtual torsion and bond angles $\kappa, \alpha \in \mathbb{R}^4$ defined over $C_\alpha$ atoms.

- Feature $\vec{r}$ between nodes representing the displacement vector between nodes.

During each training or inference stage, the energy-driven radius sampler establishes the adjacency relationship $E$ between nodes. Subsequently, it encapsulates the aforementioned protein features and inputs them into the $k$th layer of the equivariant transformer block to derive the feature matrix $Tk$. This $Tk$ matrix undergoes a high-tensor leaky operation to generate the input $Xk^{in}$ in the $k$th layer of the equivariant-elastic-SSM block. The final step involves concatenating the output of the multi-layer SSM block $\{y_k | k = 1, 2, ..., N\}$ with the output of the last layer of the transformer block $Tk$, followed by mapping them into the final node output matrix $H_{out}$.

## 4 EXPERIMENT

In this section, we present two Alphafold-predicted datasets and three experimental datasets to evaluate the performance and noise tolerance of $E^3$former. Further details are available in Appendix C.1.

### 4.1 DATASET

**CATH** We provide the dataset derived from CATH 4.2 (Ingraham et al., 2019), in which all protein structures with 40% nonredudancy are partitioned by their CATH (class, architecture, topology/fold, homologous superfamily) and kept the same data settings as a benchmark of protein representation learning (Jamasb et al., 2024). These data are split based on random assignment of the CATH topology classifications based on an 80/10/10 split.

**CATH-AF** Based on the above CATH data, we employed AlphaFold2 to predict and substitute the atomic coordinates with the predicted results, more details are illustrated in Appendix A.1.

Both CATH and CATH-AF will be utilized to evaluate in inverse folding task, a crucial step in protein design process (Dauparas et al., 2022).inverse folding involves predicting the amino acid sequence that will fold into given protein structure.

**PPBS** We provide the dataset derived from PPBS (prediction of protein–protein binding sites dataset) curated by ScanNet (Tubiana et al., 2022), which constructed a nonredundant dataset of 20K representative protein chains with annotated binding sites derived from the Dockground database of protein complexes (Kundrotas et al., 2018).

**PPBS-AF** Based on the above PPBS data, we employed AlphaFold2 to predict and substitute the atomic coordinates with the predicted results. Details about data process is provided in A.1.

Both PPBS and PPBS-AF will be utilized to evaluate in binding site prediction task. Predicting protein–protein binding sites involves identifying the residues directly involved in one or more native, high affinity PPIs. Understanding a protein's PPBS can guide docking algorithms and provide valuable insights into its in vivo behavior when its partners are unknown.

**MASIF-SITE** We utilize the experimental structures dataset sourced from the PDB by Gainza et al. (2020) and maintain the original splits. Following the pipelines described by (Jamasb et al., 2024), we label based on inter-atomic proximity (3.5 Å).

### 4.2 ALPHAFOLD-PREDICTED DATASET TASKS

We compare $E^3$former with state-of-the-arts baselines about representation learning of proteins at different levels of structural granularity ($C_\alpha$, backbones, sidechain), including SchNet (Schütt et al., 2018), TFN (Thomas et al., 2018), EGNN (Satorras et al., 2021), Equiformer (Liao & Smidt, 2022),

Table 1: The overview of tasks and and datasets.

| TASK | Structures type | Dataset Origin | #Train | #Valid | #Test | Metric |
|------|-----------------|----------------|--------|--------|-------|--------|
| Inverse Folding | Experimental | (Ingraham et al., 2019) | 18,024 | 608 | 1,120 | Perplexity |
| Binding Site Prediction | Experimental | (Tubiana et al., 2022) | 12,577 | 3,178 | 3,984 | AUPRC |
| PPI Site Prediction | Experimental | (Gainza et al., 2020) | 2,436 | 271 | 334 | AUPRC |
| Inverse Folding | AF-predicted | This work | 16,468 | 559 | 1,015 | Perplexity |
| Binding Site Prediction | AF-predicted | This work | 11,345 | 2,876 | 3,636 | AUPRC |

Table 2: Comparing different models for structure-intensive task on Alphafold-predicted datasets, - denote runs that did not converge. The decimals in the subscript represent the experimental variance. PPBS-AF All represents the combined set of all other data partitions within PPBS-AF.

| | Features | SchNet | TFN | EGNN | Equiformer | GearNet | GCPNet | $E^3$**former** |
|---|----------|--------|-----|------|------------|---------|--------|-----------|
| CATH-AF($\downarrow$) | +Seq | $9.80_{.09}$ | $7.30_{.04}$ | $8.32_{.05}$ | $6.09_{.04}$ | - | $6.20_{.08}$ | $\mathbf{5.50_{.05}}$ |
| | +$\kappa, \alpha$ | $8.71_{.07}$ | $7.28_{.04}$ | $7.72_{.06}$ | $5.62_{.03}$ | - | $6.19_{.09}$ | $\mathbf{5.44_{.04}}$ |
| | +$\phi, \psi, \omega$ | $7.32_{.08}$ | $5.11_{.02}$ | $6.07_{.05}$ | $3.64_{.03}$ | - | $3.91_{.07}$ | $\mathbf{3.54_{.04}}$ |
| PPBS-AF($\uparrow$) 70 | +Seq | $0.520_{.01}$ | $0.507_{.01}$ | $0.526_{.00}$ | $0.596_{.02}$ | $0.597_{.02}$ | $0.554_{.02}$ | $\mathbf{0.607_{.01}}$ |
| | +$\kappa, \alpha$ | $0.532_{.02}$ | $0.574_{.01}$ | $0.557_{.00}$ | $0.573_{.00}$ | $\mathbf{0.607_{.02}}$ | $0.567_{.00}$ | $0.560_{.00}$ |
| | +$\phi, \psi, \omega$ | $0.554_{.01}$ | $0.591_{.01}$ | $0.569_{.01}$ | $0.592_{.01}$ | $0.591_{.01}$ | $0.549_{.02}$ | $\mathbf{0.598_{.01}}$ |
| PPBS-AF($\uparrow$) Homology | +Seq | $0.398_{.03}$ | $0.487_{.01}$ | $0.483_{.00}$ | $0.537_{.00}$ | $0.538_{.02}$ | $0.422_{.03}$ | $\mathbf{0.546_{.01}}$ |
| | +$\kappa, \alpha$ | $0.403_{.00}$ | $0.507_{.02}$ | $0.503_{.04}$ | $\mathbf{0.547_{.03}}$ | $0.536_{.02}$ | $0.419_{.01}$ | $0.525_{.01}$ |
| | +$\phi, \psi, \omega$ | $0.428_{.01}$ | $0.516_{.00}$ | $0.518_{.00}$ | $0.547_{.04}$ | $0.541_{.00}$ | $0.425_{.02}$ | $\mathbf{0.551_{.02}}$ |
| PPBS-AF($\uparrow$) Topology | +Seq | $0.408_{.02}$ | $0.547_{.00}$ | $0.570_{.00}$ | $0.595_{.01}$ | $0.520_{.00}$ | $0.403_{.03}$ | $\mathbf{0.599_{.00}}$ |
| | +$\kappa, \alpha$ | $0.410_{.03}$ | $0.566_{.00}$ | $0.574_{.00}$ | $0.590_{.00}$ | $0.530_{.00}$ | $0.411_{.00}$ | $\mathbf{0.598_{.01}}$ |
| | +$\phi, \psi, \omega$ | $0.416_{.01}$ | $0.542_{.00}$ | $0.602_{.02}$ | $0.588_{.00}$ | $0.525_{.01}$ | $0.406_{.01}$ | $\mathbf{0.602_{.00}}$ |
| PPBS-AF($\uparrow$) None | +Seq | $0.269_{.00}$ | $0.383_{.01}$ | $0.392_{.00}$ | $0.394_{.00}$ | $0.366_{.00}$ | $0.269_{.00}$ | $\mathbf{0.405_{.01}}$ |
| | +$\kappa, \alpha$ | $0.264_{.00}$ | $0.371_{.00}$ | $0.384_{.00}$ | $0.385_{.00}$ | $0.357_{.00}$ | $0.263_{.00}$ | $\mathbf{0.409_{.01}}$ |
| | +$\phi, \psi, \omega$ | $0.274_{.01}$ | $0.357_{.00}$ | $0.407_{.01}$ | $0.394_{.01}$ | $0.366_{.00}$ | $0.271_{.00}$ | $\mathbf{0.412_{.00}}$ |
| PPBS-AF($\uparrow$) All | +Seq | $0.390_{.02}$ | $0.483_{.03}$ | $0.494_{.01}$ | $0.531_{.01}$ | $0.509_{.01}$ | $0.399_{.02}$ | $\mathbf{0.541_{.01}}$ |
| | +$\kappa, \alpha$ | $0.390_{.02}$ | $0.503_{.03}$ | $0.504_{.01}$ | $0.520_{.01}$ | $0.509_{.02}$ | $0.401_{.03}$ | $\mathbf{0.525_{.03}}$ |
| | +$\phi, \psi, \omega$ | $0.410_{.03}$ | $0.504_{.01}$ | $0.526_{.03}$ | $0.533_{.00}$ | $0.513_{.03}$ | $0.406_{.02}$ | $\mathbf{0.541_{.02}}$ |

GearNet (Zhang et al., 2022), and GCPNet (Morehead & Cheng, 2024). Most of them have been mentioned in a protein representation benchmark (Jamasb et al., 2024) and commonly employed in protein datasets. Some, like equiformer, have demonstrated good performance in 3D atom systems.

To comprehensively evaluate the performance and robustness, we conduct a comparison experiment on an AlphaFold-predicted dataset. In the inverse folding task, we utilize perplexity as the evaluation metric (Jing et al., 2020). In the binding site prediction task, we employ AUPRC due to label imbalances.

As illustrated in Table 2, $E^3$former outperforms other baselines by over 6% on average in the inverse folding task and by over 6% in the binding site prediction-PPBS-none task. Besides, at the feature level, $E^3$former demonstrates the most substantial enhancement over the baselines(11% on inverse folding task) when only Seq features are incorporated. This improvement is attributed to the increase in prediction difficulty for each model as the number of available features decreases. Similarly, $E^3$former demonstrates strong performance in the PPBS-AF-None dataset, which has a lower similarity to the training set. This difference highlights the model's ability to learn more complex interactions between amino acids with a high signal-to-noise ratio, showcasing its generalizability in difficult samples and tolerance to data deviation and noise.

### 4.3 EXPERIMENTAL DATASET TASKS

Following the experimental settings in Section 4.2, we further evaluate the model on experimental datasets. As shown in Table 3, although we replace the predicted dataset with experimental dataset, $E^3$former also leverages the inherent noise in crystal structures to enhance its performance in the comparison. The model outperforms all baseline models in datasets in most structural information intensive dataset, showing a 4% improvement in AUPRC performance in PPBS-Topology. Additionally, it consistently exhibits performance gains of over 3.5% in the inverse folding task. It is worth noting that the additional $\kappa, \alpha$ features may sometimes result in a performance decrease for each

Table 3: Comparing different models for structure-intensive task on Experimental datasets.

| | Features | SchNet | TFN | EGNN | Equiformer | GearNet | GCPNet | $E^3$**former** |
|---|---|---|---|---|---|---|---|---|
| CATH($\downarrow$) | +Seq | $11.78_{.08}$ | $10.34_{.03}$ | $10.28_{.04}$ | $7.80_{.03}$ | $12.79_{.17}$ | $8.35_{.08}$ | $\mathbf{7.53_{.02}}$ |
| | $+\kappa,\alpha$ | $11.03_{.03}$ | $10.02_{.05}$ | $9.84_{.07}$ | $8.09_{.00}$ | $12.35_{.05}$ | $8.80_{.09}$ | $\mathbf{7.97_{.02}}$ |
| | $+\phi,\psi,\omega$ | $9.97_{.09}$ | $8.73_{.02}$ | $8.89_{.04}$ | $6.91_{.04}$ | $11.61_{.12}$ | $7.56_{.11}$ | $\mathbf{6.64_{.03}}$ |
| PPBS($\uparrow$) 70 | +Seq | $0.648_{.01}$ | $\mathbf{0.781_{.01}}$ | $0.721_{.00}$ | $0.766_{.01}$ | $0.768_{.01}$ | $0.741_{.02}$ | $0.777_{.01}$ |
| | $+\kappa,\alpha$ | $0.660_{.00}$ | $\mathbf{0.783_{.00}}$ | $0.739_{.01}$ | $0.771_{.03}$ | $0.765_{.03}$ | $0.739_{.00}$ | $0.775_{.01}$ |
| | $+\phi,\psi,\omega$ | $0.658_{.01}$ | $0.783_{.02}$ | $0.752_{.02}$ | $0.780_{.02}$ | $\mathbf{0.787_{.02}}$ | $0.747_{.01}$ | $0.783_{.01}$ |
| PPBS($\uparrow$) Homology | +Seq | $0.552_{.03}$ | $0.674_{.01}$ | $0.680_{.02}$ | $0.724_{.02}$ | $0.688_{.00}$ | $0.692_{.03}$ | $\mathbf{0.732_{.02}}$ |
| | $+\kappa,\alpha$ | $0.569_{.00}$ | $0.697_{.02}$ | $0.690_{.01}$ | $0.721_{.00}$ | $0.688_{.01}$ | $0.691_{.02}$ | $\mathbf{0.727_{.02}}$ |
| | $+\phi,\psi,\omega$ | $0.568_{.03}$ | $0.694_{.00}$ | $0.702_{.02}$ | $0.727_{.03}$ | $0.692_{.01}$ | $0.702_{.02}$ | $\mathbf{0.732_{.01}}$ |
| PPBS($\uparrow$) Topology | +Seq | $0.532_{.03}$ | $0.651_{.03}$ | $0.716_{.02}$ | $0.725_{.01}$ | $0.636_{.00}$ | $0.713_{.01}$ | $\mathbf{0.743_{.02}}$ |
| | $+\kappa,\alpha$ | $0.530_{.02}$ | $0.675_{.00}$ | $0.718_{.03}$ | $0.716_{.02}$ | $0.642_{.01}$ | $0.716_{.03}$ | $\mathbf{0.745_{.00}}$ |
| | $+\phi,\psi,\omega$ | $0.519_{.01}$ | $0.668_{.00}$ | $0.733_{.02}$ | $0.728_{.01}$ | $0.640_{.03}$ | $0.721_{.02}$ | $\mathbf{0.743_{.03}}$ |
| PPBS($\uparrow$) None | +Seq | $0.455_{.02}$ | $0.542_{.04}$ | $0.596_{.05}$ | $0.618_{.04}$ | $0.540_{.01}$ | $0.588_{.04}$ | $\mathbf{0.637_{.03}}$ |
| | $+\kappa,\alpha$ | $0.464_{.01}$ | $0.567_{.04}$ | $0.602_{.04}$ | $0.604_{.04}$ | $0.540_{.03}$ | $0.581_{.00}$ | $\mathbf{0.628_{.02}}$ |
| | $+\phi,\psi,\omega$ | $0.448_{.03}$ | $0.564_{.02}$ | $0.598_{.01}$ | $0.615_{.02}$ | $0.548_{.03}$ | $0.589_{.00}$ | $\mathbf{0.631_{.01}}$ |
| PPBS($\uparrow$) All | +Seq | $0.537_{.02}$ | $0.654_{.01}$ | $0.674_{.04}$ | $0.708_{.03}$ | $0.656_{.00}$ | $0.676_{.02}$ | $\mathbf{0.724_{.02}}$ |
| | $+\kappa,\alpha$ | $0.546_{.02}$ | $0.674_{.00}$ | $0.684_{.03}$ | $0.703_{.01}$ | $0.658_{.01}$ | $0.669_{.04}$ | $\mathbf{0.719_{.02}}$ |
| | $+\phi,\psi,\omega$ | $0.539_{.00}$ | $0.672_{.00}$ | $0.692_{.01}$ | $0.712_{.03}$ | $0.660_{.01}$ | $0.685_{.01}$ | $\mathbf{0.726_{.02}}$ |
| MASIF($\uparrow$) | +Seq | $0.953_{.00}$ | $\mathbf{0.968_{.00}}$ | $0.962_{.00}$ | $0.966_{.00}$ | $0.956_{.00}$ | $\mathbf{0.968_{.00}}$ | $\mathbf{0.968_{.00}}$ |
| | $+\kappa,\alpha$ | $0.953_{.00}$ | $0.966_{.00}$ | $0.965_{.00}$ | $0.967_{.00}$ | $0.958_{.00}$ | $0.966_{.00}$ | $\mathbf{0.968_{.00}}$ |
| | $+\phi,\psi,\omega$ | $0.954_{.00}$ | $0.967_{.00}$ | $0.964_{.00}$ | $0.967_{.00}$ | $0.954_{.01}$ | $0.967_{.00}$ | $\mathbf{0.968_{.00}}$ |

model, as observed in the benchmark (Jamasb et al., 2024). This effect persists even after accounting for experimental randomness and parameter influences. We attribute this to potential information redundancy with the three-dimensional atom coordinates.

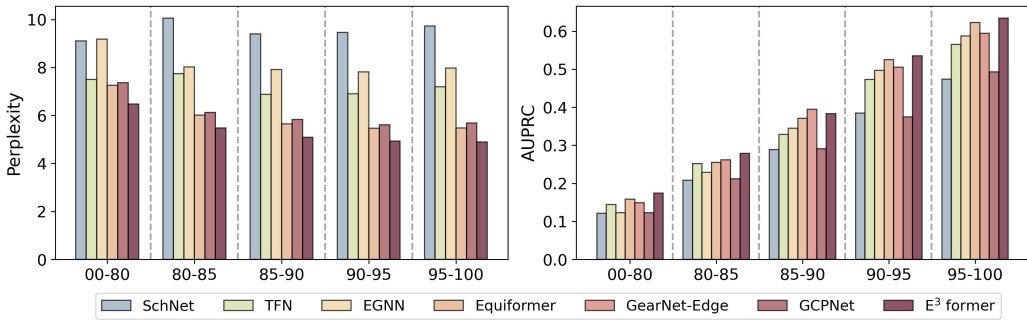

Figure 3: Comparing different models on different Alphafold-predicted confidence settings, higher confidence levels indicate that AlphaFold is more confident to its predictions accuracy. Left: CATH-AF dataset (perplexity$\downarrow$), Right: PPBS-AF All dataset (AUPRC$\uparrow$).

### 4.4 NOISE TOLERANCE EVALUATION

In order to further evaluate our model's tolerance to noise and deviation, we conducted fine-grained analysis on Alphafold-predicted and experimental data separately. First, the test set is split and compared based on AlphaFold's predicted confidence. As illustrated in Figure 3, our model demonstrates a more substantial performance enhancement when the confidence is reduced.

Similarly, we divided the experimental test set based on the structural resolution provided by RCSB (Burley et al., 2019). As shown in Figure 4, $E^3$former demonstrates competitive outcomes with low resolution dataset. These experiments collectively indicate that our model has acquired a robust representation.

### 4.5 ABLATION STUDY

In this section, ablation experiments are conducted to investigate the impact of core modules on the $E^3$former. Two modules are disabled individually: Replacing the Energy-aware protein graph module

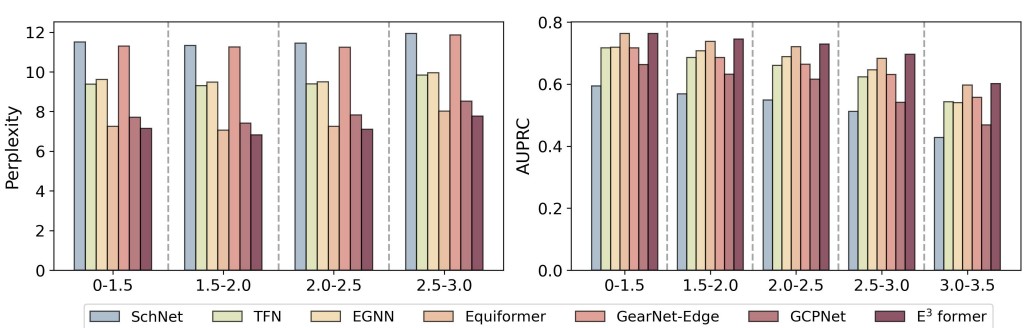

Figure 4: Comparing different on different resolution settings, a value of 0 indicates no resolution error, while a value of 3 represents a resolution error of 3Å. Left: CATH dataset (perplexity↓), Right: PPBS All dataset (AUPRC↑).

with a local radius cut-off using a $k$-nearest neighbors proximity graph. Removing the Equivariant high-tensor-elastic selective SSM module and simply employing an equivariant transformer architecture. Ablation experiments will be performed on both Alphafold-predicted datasets and additional experiments are provided in Appendix C.3. The results presented in Table 5 demonstrate that removing any core module will leads to a significant decrease in model performance. In CATH-AF, which partially relies on locality assumptions, eliminating the Energy module has a more significant impact on the model, while the Elastic module brings more stable improvements to the model. Only in rare cases (homology, $+\kappa, \alpha$), there has been a slight decline in our module's performance, possibly due to the feature introducing redundant information that interferes with the model's training.

Table 4: Ablation studies for key components in Alphafold-predicted dataset.

| Method | Features | CATH-AF($\downarrow$) | PPBS-AF($\uparrow$) | | | | |
|---|---|---|---|---|---|---|---|
| | | | 70 | Homology | Topology | None | All |
| $E^3$former | $+$Seq | $\mathbf{5.50_{05}}$ | $\mathbf{0.607_{01}}$ | $\mathbf{0.546_{01}}$ | $\mathbf{0.599_{00}}$ | $0.405_{01}$ | $\mathbf{0.541_{01}}$ |
| | $+\kappa, \alpha$ | $\mathbf{5.44_{04}}$ | $0.560_{00}$ | $0.525_{01}$ | $\mathbf{0.598_{01}}$ | $0.409_{01}$ | $0.525_{03}$ |
| | $+\phi, \psi, \omega$ | $\mathbf{3.54_{04}}$ | $0.598_{01}$ | $0.551_{02}$ | $\mathbf{0.602_{00}}$ | $0.412_{00}$ | $\mathbf{0.541_{02}}$ |
| w/o Energy | $+$Seq | $5.82_{06}$ | $0.603_{02}$ | $0.542_{03}$ | $0.597_{01}$ | $0.399_{02}$ | $0.533_{03}$ |
| | $+\kappa, \alpha$ | $5.61_{03}$ | $0.571_{01}$ | $0.528_{02}$ | $0.593_{03}$ | $0.402_{01}$ | $0.523_{02}$ |
| | $+\phi, \psi, \omega$ | $3.58_{03}$ | $0.597_{02}$ | $0.549_{01}$ | $0.598_{02}$ | $0.404_{01}$ | $0.536_{03}$ |
| w/o Elastic | $+$Seq | $5.71_{04}$ | $0.601_{02}$ | $0.543_{03}$ | $0.596_{01}$ | $0.403_{02}$ | $0.536_{02}$ |
| | $+\kappa, \alpha$ | $5.52_{04}$ | $0.563_{01}$ | $0.541_{02}$ | $0.596_{03}$ | $0.405_{01}$ | $0.525_{02}$ |
| | $+\phi, \psi, \omega$ | $\mathbf{3.54_{02}}$ | $0.593_{02}$ | $0.549_{01}$ | $0.595_{02}$ | $0.407_{02}$ | $0.539_{03}$ |
| w/o EE | $+$Seq | $6.09_{01}$ | $0.596_{02}$ | $0.537_{03}$ | $0.595_{01}$ | $0.394_{02}$ | $0.531_{02}$ |
| | $+\kappa, \alpha$ | $5.62_{05}$ | $\mathbf{0.573_{01}}$ | $\mathbf{0.547_{02}}$ | $0.590_{03}$ | $0.385_{01}$ | $0.520_{02}$ |
| | $+\phi, \psi, \omega$ | $3.64_{05}$ | $0.592_{02}$ | $0.547_{01}$ | $0.588_{02}$ | $0.394_{01}$ | $0.533_{03}$ |

## 5 CONCLUSION AND LIMITATION

In this work, we proposed an Energy-aware Elastic Equivariant Transformer-SSM hybrid architecture for 3D macromolecular structure representation learning. The core of the model is to adaptively utilize the energy-aware module to bulid proximity graph and use the equivariant SSM module to express high-order features sparsely. The above improvements enable model more tolerant to data deviation and noise. The experimental results demonstrate the superior performance of $E3$former on inverse folding, binding sites prediction, and protein-protein interaction tasks.

**Limitation** $E^3$former mainly performing operations on the 3D atom system derived from protein structures, neglecting the comprehensive utilization of others modal information (such as sequential information, chemical bond specifics, protein functional annotations, etc.), which may limits the performance of the method.

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

## A  EXPERIMENTAL DETAILS

### A.1  DETAILS OF DATASET

**CATH** The CATH dataset is a collection of protein structures curated by (Ingraham et al., 2019). In the CATH 4.2 40% non-redundant set of proteins, only chains up to a length of 500 are retained. Any chains from test set with CATH topology classifications overlap(CAT) with train are excluded. Consequently, the training, validation, and test splits consist of 18204, 608, and 1120 structures, respectively. CATH or their Alphafold-predicted version are utilized to Inverse folding task. In this node classification task, the model is trained to learn a mapping function for each residue to an amino acid type $y \in \{1, ..., n\}$, where the vocabulary size is $n = 20$, representing the 20 common amino acids.

**PPBS** We utilized the PPBS dataset as compiled by (Tubiana et al., 2022) following the data division method they outlined. Based on the Dockground database of protein-protein interfaces (Kundrotas et al., 2018), each unique PDB chain involved in one or more interfaces is considered a single example. Chains with a sequence length of less than 10 or involved in designed proteins are excluded from the dataset.In particular, the validation set and test set in PPBS are divided according to the following criteria:

- **70%**: at least 70% sequence similarity to one sample in the training set.

- **Homology**: at most 70% sequence similarity with any sample in the training set and belong to the same protein superfamily with at least one training sample.

- **Topology**: sharing a similar topological structure (T level of CATH classification (Sillitoe et al., 2021)) with at least one training sample and do not belong to the 70% dataset or the Homology dataset.

- **None**: none of the above.

- **All**: the combination of the divisions mentioned above.

In this node-level binary classification task, the model is trained to learn a mapping function for each residue to $0/1$ types, in order to discover potential protein-protein interfaces between the target protein and other proteins.

Table 5: Overviews of PPBS data partitions.

| Split type | Structures | Description |
|:---:|:---:|:---:|
| 70% | 554 | Seq.identity |
| Homology | 1485 | same superfamliy |
| Topology | 915 | similar topology |
| None | 1077 | none of above |
| all | 4031 | all of above |

**MASIF-SITE** is a dataset proposed by (Gainza et al., 2020), sourced from the PRISM list of nonredundant proteins (Baspinar et al., 2014), the ZDock benchmark (Liu et al., 2015), and SabDab (Dunbar et al., 2014). To ensure a fair comparison within the benchmark framework, we adopted the data labeling methodology outlined by (Jamasb et al., 2024) and set the binding threshold at 3.5Å. Since the labels of the corresponding Alphafold predicted version are difficult to align with the original dataset, only original dataset utilized for the experiments. The primary challenge in mapping the MASIF dataset to AlphaFold-predicted structures arose during sequence alignment using the BLOSUM62 global alignment algorithm. We observed that the resulting alignments were highly fragmented, resulting in short, disconnected segments rather than continuous regions. Given these difficulties in obtaining reliable results, we made the decision to exclude the AlphaFold-predicted dataset from our benchmark experiments.

This dataset is used to evaluate protein–protein interaction site prediction in protein surfaces, and in this node-level binary classification task, models are trained to map each residue into 0/1 type for detecting the PPI sites.

**CATH-AF** The CATH-AF dataset is created by mapping CATH entries to UniProt IDs within the AlphaFold Protein Structure Database [1]. Since the original CATH dataset only includes index information as PDB IDs, we used the PDBe REST API[2] to convert these PDB IDs into the UniProt IDs required by the AlphaFold database. All predicted protein structures are retrieved from the AlphaFold V4 protein structure database. However, some PDB IDs could not be matched to UniProt IDs during the conversion process, and due to limitations of the AlphaFold model[3], predictions for certain proteins could not be generated. Consequently, the CATH-AF dataset contains fewer structures than the original CATH dataset, with 16,468 files in the training set, 559 in the validation set, and 1,015 in the test set. Additionally, the CATH-AF dataset is proposed as a resource for protein inverse folding tasks, particularly for evaluating the protein design capabilities of models using AlphaFold-predicted structures.

**PPBS-AF** We construct a new dataset based on the PPBS dataset using structures predicted by AlphaFold. Following a similar approach to that used in CATH-AF, we employ the PDBe REST API to convert the data into the UniProt IDs required by the AlphaFold database. With pre-calculated protein binding sites provided by (Kundrotas et al., 2018), we apply the BLOSUM62 global alignment algorithm (Styczynski et al., 2008) to align the sequences of the AlphaFold-predicted structures with those of the corresponding experimental structures. This allows us to map the active sites from the reference structures onto the predicted structures. After excluding protein chains for which predicted structures are unavailable, we ultimately obtain a training set with 11,345 structures, a validation set with 2,876 structures, and a test set with 3,636 structures.

### A.2 TRAINING AND HYPERPARAMETERS

To train the model for the inverse folding task (CATH, CATH-AF), we use cross entropy Loss:

$$\mathcal{L} = -\sum_{i=1}^{K} y_i \log(\hat{y}_i)$$

To train the model for the binding sites prediction(PPBS, PPBS-AF) and Protein Protein Interaction(MASIF-SITE) tasks, we used BCE Loss function:

$$\mathcal{L} = -y \log \hat{y} - (1 - y) \log(1 - \hat{y})$$

We used the same parameters for all the above tasks. We set the maximum epochs as 50 and employed early stopping based on the validation set performance. For the $E^3$former, the hyperparameters in the Energy-Aware Protein Graph 3.1 in all tasks are set to $\epsilon = 1.0, \sigma = 3.8, \alpha = 1, \beta = 8$ to reflect the model's robustness to hyperparameters. For transformer blocks, we set the number of blocks as 6 and only keep the tensor with $L_{max} = 2$ as the input of the next block. For equivariant elastic SSM blocks, we extracted the high-order tensor of $L_3 \sim L_5$ from the upper transformer blocks as input and set the output tensor to the scalar $L = 0$. For other general settings, we set the parameters of all models according to the default configuration on a protein benchmark (Jamasb et al., 2024). Learning rate is set to 0.001, batch size is set to 16 or 32, dropout is set to 0, and a unified output head with the decoder consisting of two layers of 128 hidden units.

## B PROPERTY OF $E^3$FORMER

### B.1 APPROXIMATION OF ROTATION AND TRANSLATION NOISE

**Description.** Suppose an input vector $X$ is subject to rotation and translation noise, denoted as $\sigma_r$ and $\sigma_t$ respectively. Deep tensor product is defined as:

$$h_{m_3}^{(L_3)} = \sum_{m_1=-L_1}^{L_1} \sum_{m_2=-L_2}^{L_2} C_{(L_1,m_1)(L_2,m_2)}^{(L_3,m_3)} f_{m_1}^{(L_1)} g_{m_2}^{(L_2)},$$

---

[1]https://alphafold.ebi.ac.uk
[2]https://www.ebi.ac.uk/pdbe/api/doc/sifts.html
[3]https://alphafold.ebi.ac.uk/faq

Here, $C$ is the Clebsch-Gordan coefficient. Type-0 tensors are invariant to rotation groups and solely influenced by translations and are only affected by translations.Considering the following two network structures:

- **Structure 1**: Utilizes distinct learnable weight matrices $W_1^T, ..., W_N^T$ for continuous deep tensor products.

- **Structure 2**: Employs a singular shared learnable weight matrix $W^H$ for deep tensor product, retaining only Type-0 tensors.

We will now analyze and compare how input noise affects the output in these two different structures.

**Theorem.** *For the angle perturbation $\delta_\theta$, the corresponding rotation matrix is denoted as $R(\delta_\theta)$, satisfying $\|R(\delta_\theta) - I\| \leq \sigma_r$. For the translation noise, represented by the vector $\delta_\theta$, satisfying $\|R(\delta_\theta) - I\| \leq \sigma_t$. The structure 1 noise $\delta Y^{s1}$ satisfies $\|\delta Y^{s1}\| \leq \|W_N^T\|...\|W_1^T\|(\sigma_r\|X\| + \sigma_t)$, the structure 2 noise $\delta Y^{s2}$ satisfies $\|\delta Y^{s2}\| \leq C\sigma_t$ where $C$ is a constant.*

*Proof.* For $Z_1 = f(W_{T1}, \tilde{X})$, the noise of structure 1 can be expressed as:

$$\delta Z_1 = f(W_{T1}, \tilde{X}) - f(W_{T1}, X) = W_{T1}(R(\delta\theta)X + \delta t - X),$$

$$R(\delta\theta) \approx I + [\delta\theta] * \times,$$

where $[\delta\theta] * \times$ is an antisymmetric matrix, so we can approximate that:

$$\delta Z_1 \approx W_{T1}([\delta\theta]_\times X + \delta t).$$

Finally, the noise of Structure 1 can be expressed as:

$$\|\delta Y_{S1}\| \leq \|W_{T_N}\|\|\delta Z_{N-1}\| \leq \|W_{T_N}\|...\|W_{T_1}\|(\sigma_r\|X\| + \sigma_t).$$

For the noise of Structure 2, we have:

$$Z_1 = L_{(0,0)}^{leaky}[f(W_H, \tilde{X})].$$

Since only type-0 tensors are reserved, the impact of rotation noise on $Z$ can be ignored, and the effect of translation noise is:

$$\delta Z_1 = L_{(0,0)}^{leaky}[f(W_H, \delta t)],$$

so we have $\|\delta Z_0\| \leq C_1\sigma_t$, where $C_1$ is a constant related to $W_H$.

Similarly:

$$Z_N = L_{(0,0)}^{leaky}[f(W_H, Z_{N-1})],$$

$$\|\delta Y_{S2}\| \leq C_N\|\delta Z_{N-1}\| \leq C_N...C_1\sigma_t.$$

For the noise of structure 2, we have: $Z_1 = L_{(0,0)}^{leaky}[f(W_H, \tilde{X})]$. Since only Type-0 tensors are retained, the effect of rotation noise on Z can be ignored, and the effect of translation noise is:

$$\delta Z_1 = L_{(0,0)}^{leaky}[f(W_H, \delta t)],$$

so we have $\|\delta Z_0\| \leq C_1\sigma_t$, where $C_1$ is a constant related to $W_H$. Similarly:

$$Z_N = L_{(0,0)}^{leaky}[f(W_H, Z_{N-1})], \|\delta Y_{S2}\| \leq C_N\|\delta Z_{N-1}\| \leq C_N...C_1\sigma_t.$$

In Structure 1, the noise is amplified by a factor of $\|W_{T_N}\|...\|W_1\|$, and the rotation noise $\sigma_r$ significantly affects the output.

In Structure 2, since only Type-0 tensors are retained each step and the impact of rotation noise $\sigma_r$ is minimal, the effect of translation noise is also limited to a constant multiple $C$. Consequently, the Equivariant Elastic High-tensor leaky SSM architecture can effectively reduce the impact of noise on the output.

Table 6: Comparing different models for structure-intensive task on Alphafold-predicted datasets under different confidence cutoffs(0-80%/80-85%/85-90%/90-95%/95-100%), highlighting denotes the best performance among all compared methods under different confidence interval.

| | Features | SchNet | TFN | EGNN | Equiformer | GearNet | $E^3$former |
|---|---|---|---|---|---|---|---|
| CATH-AF(↓) | +Seq | 9.11/10.05/9.40/9.46/9.72 | 7.50/7.74/6.87/6.91/7.20 | 9.18/8.02/7.91/7.81/7.97 | 7.26/6.01/5.65/5.47/5.48 | 16.29/38.54/-/34.16/42.03 | 6.48 / 5.48 / 5.09 / 4.93 / 4.90 |
| | +κ,α | 8.81/8.98/8.59/8.43/8.82 | 7.31/7.39/7.08/6.99/7.33 | 8.40/7.85/7.33/7.32/7.41 | 6.54 / 5.53 /5.23/5.08/5.07 | 12.11/18.81/18.31/18.99/22.87 | 6.63/5.61/ 5.18 / 5.06 / 5.07 |
| | +φ,ψ,ω | 7.57/7.62/7.14/7.02/7.51 | 5.18/5.73/4.87/4.96/5.23 | 6.37/6.22/5.86/5.84/5.91 | 4.10/3.69/3.43/3.34/3.38 | 8.00/9.45/8.63/8.98/9.89 | 4.01 / 3.54 / 3.29 / 3.23 / 3.28 |
| PPBS-AF(↑) 70 | +Seq | 0.292/-/0.374/0.453/0.618 | 0.266/-/0.364/0.436/0.596 | 0.269/-/0.397/0.511/0.640 | 0.301/-/0.450/0.556/0.674 | 0.343/-/ 0.509 / 0.598 / 0.703 | 0.366 /-/ 0.458/ 0.582/0.678 |
| | +κ,α | 0.234/-/0.370/0.497/0.599 | 0.255/-/0.486/0.504/0.652 | 0.253/-/0.422/0.523/0.628 | 0.314/-/ 0.532 /0.546/0.653 | 0.325 /-/0.513/ 0.591 / 0.710 | 0.317/-/0.490/0.545/0.644 |
| | +φ,ψ,ω | 0.266/-/0.467/0.512/0.660 | 0.287/-/0.419/0.553/ 0.711 | 0.349 /-/0.412/0.566/0.642 | 0.329/-/0.527/0.568/0.661 | 0.320/-/ 0.589 / 0.579 /0.677 | 0.323/-/0.533/0.578/0.664 |
| PPBS-AF(↑) Homology | +Seq | 0.120/0.212/0.315/0.398/0.463 | 0.223 /0.267/0.371/0.474/0.554 | 0.130/0.225/0.369/0.489/0.569 | 0.185/0.278/0.399/0.541/0.614 | 0.118/0.324/0.421/0.539/0.602 | 0.152/ 0.340 / 0.425 / 0.544 / 0.624 |
| | +κ,α | 0.178/0.224/0.311/0.402/0.465 | 0.145/0.300/0.419/0.512/0.577 | 0.198/0.262/0.398/0.496/0.575 | 0.194/0.321/0.424/ 0.547 / 0.612 | 0.214 / 0.339 / 0.451 /0.535/0.606 | 0.186/0.328/0.411/0.530/0.603 |
| | +φ,ψ,ω | 0.104/0.264/0.328/0.429/0.493 | 0.142/0.295/0.413/0.511/0.587 | 0.189/0.278/0.397/0.515/0.583 | 0.136/0.271/0.446/ 0.555 /0.615 | 0.179/0.299/0.460/0.538/0.607 | 0.234 / 0.306 / 0.465 /0.546/ 0.620 |
| PPBS-AF(↑) Topology | +Seq | 0.058/-/0.214/0.429/0.472 | 0.084 /-/0.355/0.548/0.598 | 0.055/-/0.364/0.592/0.626 | 0.074/-/0.380/0.600/0.659 | 0.073/-/0.300/0.502/0.586 | 0.071/-/ 0.389 / 0.608 / 0.662 |
| | +κ,α | 0.063/-/0.242/0.396/0.464 | 0.053/-/0.303/0.560/0.621 | 0.057/-/0.259/0.569/0.624 | 0.055/-/0.328/0.584/0.647 | 0.088 /-/0.281/0.522/0.588 | 0.032/-/ 0.360 / 0.605 / 0.654 |
| | +φ,ψ,ω | 0.050/-/0.217/0.391/0.482 | 0.077 /-/0.294/0.528/0.594 | 0.053/-/0.381/ 0.607 /0.652 | 0.072/-/0.342/0.581/0.640 | 0.043/-/0.250/0.491/0.585 | 0.070/-/ 0.401 /0.594/ 0.654 |
| PPBS-AF(↑) None | +Seq | 0.109/0.145/0.225/0.311/0.342 | 0.116/ 0.205 /0.290/0.424/0.465 | 0.115/0.174/ 0.322 /0.460/0.493 | 0.152 /0.159/0.261/ 0.460 /0.507 | 0.106/0.152/0.296/0.411/0.473 | 0.145/0.178/0.315/0.459/ 0.513 |
| | +κ,α | 0.111/0.148/0.219/0.300/0.327 | 0.117/0.134/ 0.332 /0.434/0.486 | 0.129/0.146/0.287/0.443/0.481 | 0.149 /0.169/0.317/0.449/0.515 | 0.138/0.151/0.291/0.396/0.463 | 0.121/ 0.173 /0.298/ 0.466 / 0.523 |
| | +φ,ψ,ω | 0.105/0.132/0.234/0.330/0.349 | 0.113/0.142/0.279/0.406/0.461 | 0.125/0.169/0.264/ 0.460 / 0.516 | 0.134/0.169/0.295/0.443/0.512 | 0.113/0.155/0.291/0.414/0.488 | 0.156 / 0.180 / 0.305 /0.459/0.513 |
| PPBS-AF(↑) All | +Seq | 0.121/0.208/0.289/0.385/0.474 | 0.144/0.252/0.329/0.473/0.565 | 0.123/0.229/0.345/0.497/0.587 | 0.158/0.255/0.371/0.525/0.623 | 0.149/0.262/ 0.395 /0.505/0.594 | 0.174 / 0.279 /0.383/ 0.535 / 0.634 |
| | +κ,α | 0.125/0.215/0.279/0.380/0.469 | 0.122/0.223/0.376/0.499/0.593 | 0.139/0.223/0.364/0.497/0.590 | 0.159/ 0.265 /0.380/ 0.528 /0.622 | 0.166 /0.251/0.384/0.497/0.590 | 0.162/0.259/ 0.390 /0.521/ 0.627 |
| | +φ,ψ,ω | 0.108/0.233/0.295/0.400/0.490 | 0.135/0.253/0.379/0.494/0.596 | 0.153/0.257/0.363/0.516/0.613 | 0.156/0.257/ 0.398 /0.523/0.622 | 0.146/0.253/0.394/0.503/0.595 | 0.172 / 0.271 /0.397/ 0.534 / 0.633 |

## C ADDITIONAL RESULTS

### C.1 NOISE TOLERANCE EVALUATION

We added more details about the Noise Tolerance Evaluation experiment 4.4, which involves noise tolerance evaluation under various features and data partitions. As shown in Tables 6. $E^3$former shows more robustness in various tasks under low confidence. Methods based on invariance, such as SchNet and GearNet, also demonstrate notable noise tolerance.

### C.2 CASE STUDY

Two protein complexes with experimental resolutions of 3.5Å are provided in Figure 5. Figure (a) shows the interactions of chain A with other chains in 1KPK, while Figure (b) shows the interactions of chain E with other chains in 3DXA. This visualization demonstrates the details of $E^3$former. It can be observed that even at lower structural resolutions, our model tends to assign high prediction probabilities to the binding regions of proteins and significantly low probabilities to nodes far from these regions.

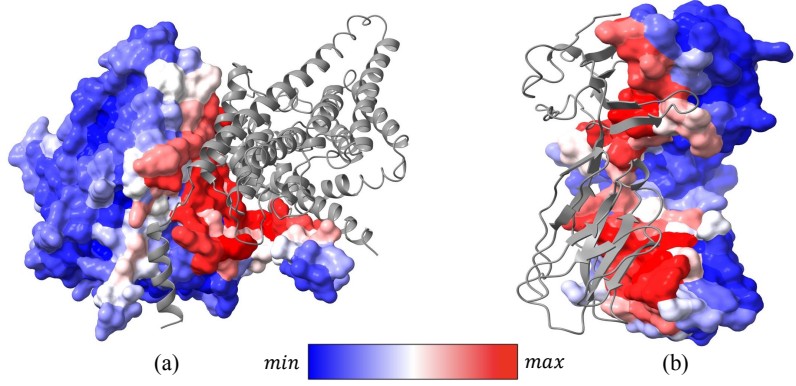

(a)    *min*    *max*    (b)

Figure 5: Visualization of protein binding sites predition. The predicted probability of amino acid binding sites is depicted using varying color intensities. (a) 1KPK (b) 3DXA

### C.3 ABLATION STUDY

As an supplement to Section 4.5, we performed ablation experiments on the experimental dataset. In comparison to the findings in the AlphaFold-predicted dataset (Table 5), the removal of the Energy module has a more pronounced effect on the outcomes than excluding the Elastic module. This is because the Elastic module tends to benefit from substantial more significant noise or data bias, while the Energy module is not only aids in learning a more robust representation, but also adaptively modeling structural data with different distributions.

Table 7: Ablation studies for key components in Experimental dataset.

| Method | Features | CATH($\downarrow$) | PPBS($\uparrow$) | | | | | MASIF($\uparrow$) |
|---|---|---|---|---|---|---|---|---|
| | | | 70 | Homology | Topology | None | All | |
| $E^3$former | +Seq | $\mathbf{7.53}_{.02}$ | $\mathbf{0.777}_{.01}$ | $\mathbf{0.732}_{.02}$ | $\mathbf{0.743}_{.02}$ | $\mathbf{0.637}_{.03}$ | $\mathbf{0.724}_{.02}$ | $\mathbf{0.968}_{.00}$ |
| | $+\kappa, \alpha$ | $\mathbf{7.97}_{.02}$ | $\mathbf{0.775}_{.01}$ | $\mathbf{0.727}_{.02}$ | $\mathbf{0.745}_{.00}$ | $\mathbf{0.628}_{.02}$ | $\mathbf{0.719}_{.02}$ | $\mathbf{0.968}_{.00}$ |
| | $+\phi, \psi, \omega$ | $\mathbf{6.64}_{.03}$ | $\mathbf{0.783}_{.01}$ | $\mathbf{0.732}_{.01}$ | $\mathbf{0.743}_{.03}$ | $\mathbf{0.631}_{.01}$ | $\mathbf{0.726}_{.02}$ | $\mathbf{0.968}_{.00}$ |
| w/o Energy | +Seq | $7.61_{.02}$ | $0.771_{.01}$ | $0.725_{.03}$ | $0.729_{.02}$ | $0.625_{.04}$ | $0.711_{.03}$ | $0.967_{.00}$ |
| | $+\kappa, \alpha$ | $8.07_{.01}$ | $0.773_{.00}$ | $0.724_{.03}$ | $0.732_{.00}$ | $0.613_{.02}$ | $0.709_{.01}$ | $0.967_{.00}$ |
| | $+\phi, \psi, \omega$ | $6.76_{.01}$ | $0.779_{.01}$ | $0.727_{.02}$ | $0.734_{.00}$ | $0.623_{.01}$ | $0.719_{.03}$ | $\mathbf{0.968}_{.00}$ |
| w/o Elastic | +Seq | $7.68_{.02}$ | $0.773_{.03}$ | $0.728_{.00}$ | $0.735_{.01}$ | $0.631_{.00}$ | $0.715_{.01}$ | $0.967_{.00}$ |
| | $+\kappa, \alpha$ | $8.03_{.00}$ | $0.772_{.00}$ | $0.723_{.00}$ | $0.729_{.02}$ | $0.619_{.03}$ | $0.712_{.00}$ | $0.967_{.03}$ |
| | $+\phi, \psi, \omega$ | $6.79_{.00}$ | $0.782_{.00}$ | $0.729_{.01}$ | $0.739_{.02}$ | $0.627_{.03}$ | $0.722_{.01}$ | $0.967_{.00}$ |
| w/o EE | +Seq | $7.80_{.00}$ | $0.765_{.00}$ | $0.722_{.00}$ | $0.723_{.02}$ | $0.615_{.00}$ | $0.703_{.02}$ | $0.966_{.00}$ |
| | $+\kappa, \alpha$ | $8.12_{.03}$ | $0.771_{.00}$ | $0.719_{.00}$ | $0.713_{.00}$ | $0.602_{.01}$ | $0.701_{.00}$ | $0.967_{.00}$ |
| | $+\phi, \psi, \omega$ | $6.89_{.01}$ | $0.778_{.02}$ | $0.726_{.01}$ | $0.725_{.03}$ | $0.617_{.00}$ | $0.711_{.02}$ | $0.967_{.00}$ |

## C.4 CONVERGENCE ANALYSIS

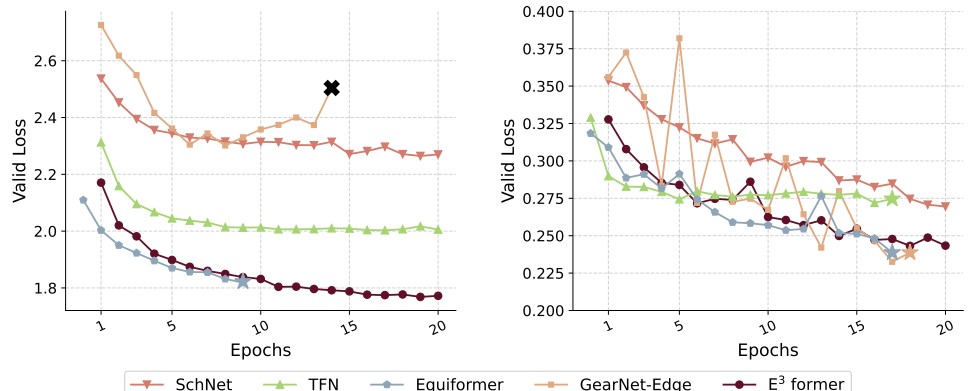

Figure 6: Validation loss values over epochs in Alphafold-predicted data, (cross mark: loss changes abnormally, star mark: early stop.

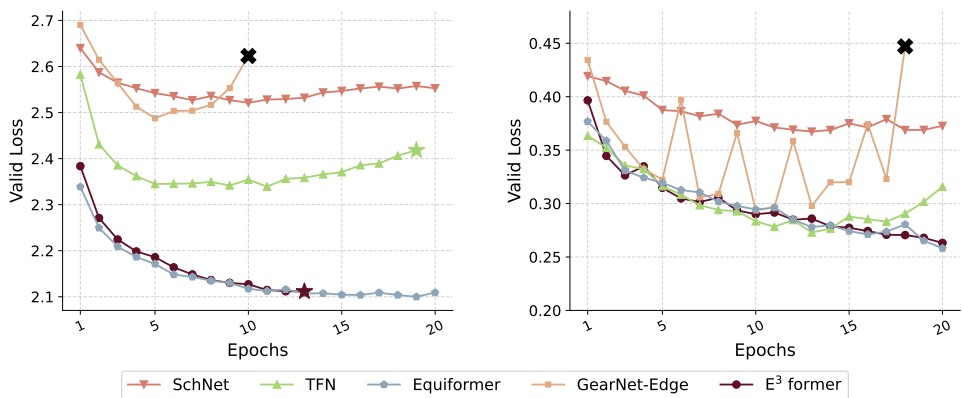

Figure 7: Validation loss values over epochs in experimental data, (cross mark: loss changes abnormally, star mark: early stop.

We conducts the convergence speeds comparison experiment of the models across various datasets. Figure 6 and Figure 7 demonstrate that all the models eventually converged, with both $E^3$former

and Equiformer achieving convergence relatively early. This rapid convergence could be attributed to their utilization of the equivariant Transformer architecture based on irreducible representations, which inherently enables fast fitting.

## D  BOARDER IMPACT

In this work, we propose an adaptive energy-aware elastic equivariant Transformer model for learning protein representations. Additionally, we introduce two datasets based on Alphafold-predicted protein structures, specifically designed to tackle challenges in protein design and function prediction in the post-Alphafold era. Our model effectively addresses the noise introduced by low-resolution in experimentally determined protein structures, as well as the systematic errors inherent in Alphafold-predicted structures.

We strongly believe that computational biology tools like Alphafold will profoundly reshape our understanding and exploration of structural biology, drug discovery, and other natural sciences in the future. Therefore, this work aims to provide new insights for protein design and drug discovery in the context of structural biology models driven by such computational tools. Developing more robust and noise-resistant protein representations will also enable better predictions of protein functions and physicochemical properties under extreme conditions, pushing the boundaries of life science research.

