# OpenReview forum: "$E^3$former: An Adaptive Energy-Aware Elastic Equivariant Transformer Model For Protein Representation Learning"
_ICLR.cc/2025/Conference — Submitted to ICLR 2025_

### Official Review · Reviewer_GWEV · 2024-10-28

**Soundness:** 2
**Presentation:** 2
**Contribution:** 2
**Rating:** 3
**Confidence:** 5

**Summary:**

The paper introduces $E^3$former, an adaptive energy-aware, elastic, equivariant Transformer model designed to handle protein representation learning effectively, especially in the context of noisy data from protein structures. Traditional models struggle with robustness due to structural noise from both experimentally determined and predicted protein models (e.g., AlphaFold structures). The E3former addresses these issues with a novel hybrid approach, combining a Transformer with a state space model (SSM) that enhances its robustness and adaptability to noise. Inspired by molecular dynamics, this model uses an energy-aware radius function and adaptive radius sampling to define local neighborhoods, mitigating data bias effects in protein proximity graphs. The model incorporates an elastic selective SSM that leverages high-order tensors and sparse representations to improve signal-to-noise ratio and retain equivariance.

**Strengths:**

1. This paper is well-organized and clearly written.
2. The combination of energy-aware radius functions and equivariant high-order tensors offers a fresh approach to handling noise in protein structures.
3. The paper exhibits a high level of technical rigor and methodological detail. Each component, from the energy-aware graph construction to the high-order tensor elastic selective SSM, is thoughtfully designed and empirically validated.

**Weaknesses:**

1. The paper’s primary motivation is to address noise in structural data from both experimental and AlphaFold-predicted sources. However, the paper lacks a detailed quantitative or qualitative analysis of the specific types and impacts of noise present in these datasets. Without a thorough analysis of how noise in these data affects current models, it is difficult to assess whether the E3former’s adaptive mechanisms effectively target the underlying issues.
2. While the authors introduce two new datasets (CATH-AF and PPBS-AF) based on AlphaFold-predicted structures, these datasets appear to be relatively simple conversions of existing datasets into AlphaFold’s predicted coordinates.
3. The experimental setup, while extensive, could be strengthened by testing E3former’s generalization across experimental and predicted data sources. For example, evaluating the model’s robustness by training on CATH-AF’s training set and then testing on the CATH test set would provide critical insights into E3former’s performance across different data domains.

**Questions:**

Can the authors provide a detailed analysis of the types and magnitudes of noise typically found in both experimental and AlphaFold-predicted protein structures? How do these impact the performance of existing models?

Did the authors test E3former’s ability to generalize by training on one data source (e.g., CATH-AF) and testing on another (e.g., experimental CATH)? If so, what were the results, and if not, what are the anticipated challenges or expected outcomes?

---

> ### Author Response · Authors · 2024-11-20
> **A kind rebuttal to Reviewer GWEV (Part 1)**
>
> **Q1:** "The paper’s primary motivation is to address noise in structural data from both experimental and AlphaFold-predicted sources. However, the paper lacks a detailed quantitative or qualitative analysis of the specific types and impacts of noise present in these datasets. Without a thorough analysis of how noise in these data affects current models, it is difficult to assess whether the E3former’s adaptive mechanisms effectively target the underlying issues."
>
> **A1:** Based on the causes and property of noise, we categorize it into three types:
>
> 1. **Perturbation Noise**: These are approximated as Gaussian-distributed noise fluctuations, primarily  caused by measurement inaccuracies of experimental instruments, molecule dynamics or environmental factors [1].
> 2. **Biased Noise in Experimental Structures**: Examples include noise introduced by resolution limitations or reconstruction algorithms in cryo-EM or NMR data [2].
> 3. **Biased Noise in AlphaFold Predictions**: These arise from structural tendencies embedded during AlphaFold’s training rather than inherent biological conformational variability. Notably, this type of noise is particularly prominent in low-confidence regions [3], and AlphaFold2 predictions often capture only one experimentally determined conformation while omitting others [4].
>
> **Perturbation noise** can be mitigated using conventional methods such as regularization and data augmentation.
>
> For **biased noise in experimental structures**, Figure 4 shows that noise in lower-resolution interval significantly impacts model performance. However, $E^{3}$former continues to demonstrate robust performance, showcasing its ability to adapt the type of noise.
>
> Similarly, for **biased noise in AlphaFold predicted data**, results in Figure 3 and Table 6 reveal that $E^{3}$former achieves more comparable performance in low-confidence regions, showcasing our model 's ability to noise tolerance.
>
> Additionally, the cross-dataset evaluations we provided in **[A3]** further quantify this issue. As the confidence of predictions decreases and the proportion of biased noise increases, $E^{3}$former exhibits a more pronounced performance improvement, reveal its effectiveness to handle different types of noise.
>
>
>
> **Q2:** "While the authors introduce two new datasets (CATH-AF and PPBS-AF) based on AlphaFold-predicted structures, these datasets appear to be relatively simple conversions of existing datasets into AlphaFold’s predicted coordinates"
>
> **A2:** Thank you for this important question, which addresses a significant challenge we encountered during data processing. To clarify, CATH-AF and PPBS-AF are not simple coordinate transformations of existing datasets. Rather, they represent careful mappings between the original protein sequence and the predicted structure corresponding to its UniProt sequence, which naturally leads to some variation in amino acid counts. While ideally AlphaFold-predicted structures should perfectly match their original protein sequences (differing only in atomic coordinates), the computational demands of folding tens of thousands of protein sequences exceeded our available resources. As detailed in the appendix, we implemented a pragmatic and scalable solution: leveraging the RCSB PDB database to map UniProt IDs to their corresponding AlphaFold Database predictions. However, this approach has a minor limitation, as the AlphaFold Database does not contain predictions for every UniProt ID.
>
> In PDB files, due to experimental conditions (e.g., X-ray crystallography, cryo-EM) or technical limitations, not all regions of a protein sequence are resolved into structures. Consequently, the sequences recorded in PDB files are typically truncated versions of the corresponding UniProt sequences. Furthermore, proteins in PDB files may have undergone experimental modifications or intentional truncation, resulting in discrepancies from their original UniProt sequences.
>
> Looking ahead, we plan to investigate open-source protein folding models with lower computational requirements (e.g., Protenix, Chai-1, Boltz-1) to enable direct structure prediction from sequences. This will allow us to evaluate model performance under conditions of greater sequence consistency.

---

> > ### Author Response · Authors · 2024-11-20
> > **A kind rebuttal to Reviewer GWEV (Part 2)**
> >
> > **Q3:** "The experimental setup, while extensive, could be strengthened by testing E3former’s generalization across experimental and predicted data sources. For example, evaluating the model’s robustness by training on CATH-AF’s training set and then testing on the CATH test set would provide critical insights into $E^{3}$former’s performance across different data domains......Did the authors test E3former’s ability to generalize by training on one data source (e.g., CATH-AF) and testing on another (e.g., experimental CATH)? If so, what were the results, and if not, what are the anticipated challenges or expected outcomes?"
> >
> > **A3:** Thank you for your insightful suggestion. We are surprised to find this additional experiment further demonstrates the model's inherent noise tolerance. We conduct additional cross-dataset evaluations to address this point. Specifically, we train $E^{3}$former on **CATH-AF (AlphaFold-predicted data)** and test it on **CATH experimental data**, and we conduct similar experiments on PPBS dataset. While some baseline methods exhibit a significant performance decline in noisy environments, our model maintained comparable performance on the test set. This demonstrates the model's robustness and ability to generalize to unseen types of noise and prediction errors. Similarly, when trained on noisy AlphaFold-predicted data and tested on experimental structures, the model also shows a significant improvement.
> >
> > |      | Features                | SchNet   | TFN      | EGNN     | Equiformer | GearNet  | GCPNet   | $E^3\text{former}$ |
> > | ---- | ----------------------- | -------- | -------- | -------- | ---------- | -------- | -------- | ------------------ |
> > | CATH | +Seq                    | 17.32792 | 14.34635 | 11.28722 | 10.96079   | 65.89864 | 11.02013 | 10.53737           |
> > |      | +$\kappa$,$\alpha$      | 15.14276 | 15.63958 | 11.15313 | 10.75782   | 28.09291 | 11.25322 | 10.33705           |
> > |      | +$\phi$,$\psi$,$\omega$ | 18.38843 | 21.74517 | 11.88394 | 11.72478   | 23.59218 | 12.63292 | 11.018949          |
> >
> >
> >
> > |               | Features                | Schnet  | TFN     | EGNN    | Equiformer | GearNet | GCPNet  | $E^3\text{former}$ |
> > | ------------- | ----------------------- | ------- | ------- | ------- | ---------- | ------- | ------- | -------------------- |
> > | PPBS 70       | +Seq                    | 0.44953 | 0.64124 | 0.62695 | 0.66591    | 0.65592 | 0.64219 | 0.68544              |
> > |               | +$\kappa$,$\alpha$      | 0.48296 | 0.65132 | 0.64405 | 0.68343    | 0.64709 | 0.63064 | 0.68185              |
> > |               | +$\phi$,$\psi$,$\omega$ | 0.52123 | 0.63546 | 0.62501 | 0.67828    | 0.64118 | 0.62339 | 0.67672              |
> > | PPBS Homology | +Seq                    | 0.37548 | 0.56347 | 0.59807 | 0.63228    | 0.61499 | 0.58236 | 0.65478              |
> > |               | +$\kappa$,$\alpha$      | 0.39545 | 0.60154 | 0.59935 | 0.64031    | 0.59546 | 0.56641 | 0.65088              |
> > |               | +$\phi$,$\psi$,$\omega$ | 0.4434  | 0.57120 | 0.58794 | 0.64042    | 0.60408 | 0.57923 | 0.65221              |
> > | PPBS Topology | +Seq                    | 0.37378 | 0.59578 | 0.65274 | 0.67387    | 0.60586 | 0.57741 | 0.69854              |
> > |               | +$\kappa$,$\alpha$      | 0.41137 | 0.65262 | 0.64796 | 0.67406    | 0.59483 | 0.61239 | 0.69284              |
> > |               | +$\phi$,$\psi$,$\omega$ | 0.43746 | 0.60515 | 0.66177 | 0.67785    | 0.60587 | 0.58901 | 0.69788              |
> > | PPBS None     | +Seq                    | 0.28093 | 0.43247 | 0.51527 | 0.51266    | 0.48786 | 0.44592 | 0.53728              |
> > |               | +$\kappa$,$\alpha$      | 0.32368 | 0.49414 | 0.51055 | 0.51803    | 0.45168 | 0.47998 | 0.53678              |
> > |               | +$\phi$,$\psi$,$\omega$ | 0.35349 | 0.45055 | 0.50075 | 0.51098    | 0.47044 | 0.43271 | 0.52693              |
> > | PPBS All      | +Seq                    | 0.35806 | 0.53397 | 0.59119 | 0.61592    | 0.58545 | 0.56649 | 0.64033              |
> > |               | +$\kappa$,$\alpha$      | 0.38917 | 0.59224 | 0.59253 | 0.62204    | 0.56501 | 0.58232 | 0.63571              |
> > |               | +$\phi$,$\psi$,$\omega$ | 0.42553 | 0.55304 | 0.58419 | 0.62065    | 0.57698 | 0.56612 | 0.63465              |
> >
> > Reference:
> >
> > [1] Pöschko, Maria Theresia, et al. "Nonlinear detection of secondary isotopic chemical shifts in NMR through spin noise." *Nature communications* 8.1 (2017): 13914.
> >
> > [2] Beton, Joseph G., et al. "Cryo-EM structure and B-factor refinement with ensemble representation." *Nature Communications* 15.1 (2024): 444.
> >
> > [3] Chakravarty, Devlina, et al. "AlphaFold predictions of fold-switched conformations are driven by structure memorization." *Nature Communications* 15.1 (2024): 7296.
> >
> > [4] Chakravarty, Devlina, and Lauren L. Porter. "AlphaFold2 fails to predict protein fold switching." *Protein Science* 31.6 (2022): e4353.

---

### Official Review · Reviewer_nr39 · 2024-11-02

**Soundness:** 3
**Presentation:** 2
**Contribution:** 3
**Rating:** 6
**Confidence:** 5

**Summary:**

The authors present E3Former, an equivariant architecture based on Transformer-State Space Model for protein representation learning. The authors show strong empirical performance on a wide range of benchmarking tasks, exploring the robustness of the proposed method to structural artefacts by examining the performance on both experimental and predicted structures. The paper is generally well written though presentation can be improved.

**Strengths:**

* Evaluating the method on both crystal and predicted structures
* Empirical results are convincing across a broad range of tasks

**Weaknesses:**

My primary concern with this work is with the authors' premise that robustness to noise is a fundamental limitation of existing works. Often, works in this area have shown that adding noise to crystal structures provides a form of regularisation that generally boosts performance -- sometimes quite substantially. I think the issue that the authors are highlighting is not so much "noise" in experimental structures, but with \emph{prediction errors} which can substantially alter the input graphs and affect model training. I'd suggest being clearer with this distinction


Also, the trend in this area of research has been towards developing large pre-trained models which can be fine-tuned for a range of tasks. I think this contribution would be strengthened by some exploration of the ability to scale the proposed method and how it compares to existing pre-trained structural encoders.


### Minor

* L49: "Highly sensitive" -> High sensitivity
* "while also tackling the flexible property of macromolecules" - What exactly are the authors referring to?
* L79: " t omaintain" -> to maintain
* L79: repetition of "in various tasks"
* Mind missing spaces thoughout (L129,132, 271, 272, 305, 307)
* L189: Should this not be \Epsilon_{norm}
* L273: Inconsistent bolding of $G$
* I believe L277 should read: "NOT used in inverse folding task"
* L283: Displacement vector?
* Table 5: seq indentity -> Seq. identity
* Appendix D: "BOARDER IMPACT" -> "Broader Impact"

**Questions:**

* How much hyperparameter tuning did the authors do? Particularly for the models that were not drawn from the ProteinWorkshop benchmark such as equiformer.


* L284: During each training or inference stage -> What does stage refer to here? Every epoch? Every batch? Or is the adjacency fixed for a given training run?
* In Figure 4, a crystallographer would consider all of these resolutions extremely good -> good. I suggest expanding the range of resolutions considered
* Were the ablation studies hyperparameter tuned?
* What does the highlighting denote in Table 6?
* In the appendix, the authors mention there were difficulties mapping the Masif site dataset labels to AF2 predictions. Could they please elaborate a little here?

---

> ### Author Response · Authors · 2024-11-20
> **A kind rebuttal to Reviewer nr39 (Part 1)**
>
> I sincerely appreciate your careful revisions and constructive feedback on this paper. I believe your suggestions are truly inspiring and will greatly help me enhance the quality of this work. I will now provide detailed responses to your questions.
>
> **Q1:** "My primary concern with this work is with the authors' premise that robustness to noise is a fundamental limitation of existing works. Often, works in this area have shown that adding noise to crystal structures provides a form of regularisation that generally boosts performance -- sometimes quite substantially. I think the issue that the authors are highlighting is not so much "noise" in experimental structures, but with \emph{prediction errors} which can substantially alter the input graphs and affect model training. I'd suggest being clearer with this distinction"
>
> **A1:** We agree that the inherent biases or systematic errors in both experimental techniques (e.g., cryo-EM or NMR)[1] and AlphaFold-predicted structures reflect non-random, structured deviations[2,3]. However, from the model's perspective, since $E^{3}$former does not introduce the above information as inductive bias, all types of deviations can be generalized as "noise" without explicit prior knowledge. This assumption allows our framework to treat these biases and errors in a uniform manner, enabling robustness without detailed priors about the specific data source or noise characteristics. To handle such diverse noise sources, $E^{3}$former introduces Molecular Dynamics Signals(Energy-aware radius graph) and utilizes high signal-to-noise ratio information extraction method(Elastic selective mechanism).  These innovations enable $E^{3}$former to generalize effectively across a wide range of noise types.
>
> To further clarify this point, we conducted additional cross-dataset evaluations: We trained $E^{3}$former on **PPBS experimental data** and tested it on **PPBS-AF (AlphaFold-predicted data)**. Despite the systematic differences and the lack of noise-specific inductive biases in the training data, the model achieved comparable performance in the test set. This highlights the model’s ability to generalize to unseen types of noise and prediction errors. Conversely, training on noisy AlphaFold-predicted data and testing on experimental structures also demonstrated strong predictive ability.
>
> |                  | Features                | Schnet  | TFN     | EGNN    | Equiformer | GearNet | GCPNet  | $E^3\text	{former}$ |
> | ---------------- | ----------------------- | ------- | ------- | ------- | ---------- | ------- | ------- | ---------------------- |
> | PPBS-AF 70       | +Seq                    | 0.20168 | 0.36930 | 0.28549 | 0.32871    | 0.35460 | 0.22138 | 0.33082                |
> |                  | +$\kappa$,$\alpha$      | 0.21044 | 0.31716 | 0.20516 | 0.33254    | 0.32297 | 0.23769 | 0.34731                |
> |                  | +$\phi$,$\psi$,$\omega$ | 0.28123 | 0.37468 | 0.26797 | 0.33381    | 0.33172 | 0.29985 | 0.34371                |
> | PPBS-AF Homology | +Seq                    | 0.14265 | 0.25322 | 0.22522 | 0.26929    | 0.23107 | 0.17286 | 0.27231                |
> |                  | +$\kappa$,$\alpha$      | 0.14966 | 0.23096 | 0.14952 | 0.26761    | 0.21692 | 0.18126 | 0.27503                |
> |                  | +$\phi$,$\psi$,$\omega$ | 0.20689 | 0.27209 | 0.20752 | 0.28433    | 0.22315 | 0.21829 | 0.29445                |
> | PPBS-AF Topology | +Seq                    | 0.22531 | 0.35238 | 0.39045 | 0.40333    | 0.30934 | 0.26523 | 0.41394                |
> |                  | +$\kappa$,$\alpha$      | 0.23966 | 0.34851 | 0.23546 | 0.39530    | 0.31184 | 0.26981 | 0.41967                |
> |                  | +$\phi$,$\psi$,$\omega$ | 0.26613 | 0.36855 | 0.26359 | 0.41227    | 0.30314 | 0.29314 | 0.42827                |
> | PPBS-AF None     | +Seq                    | 0.10728 | 0.15740 | 0.15415 | 0.15649    | 0.13569 | 0.11267 | 0.17847                |
> |                  | +$\kappa$,$\alpha$      | 0.11008 | 0.15162 | 0.10940 | 0.16689    | 0.13152 | 0.12901 | 0.16882                |
> |                  | +$\phi$,$\psi$,$\omega$ | 0.13614 | 0.17037 | 0.13929 | 0.18115    | 0.12171 | 0.12691 | 0.18519                |
> | PPBS-AF All      | +Seq                    | 0.16702 | 0.26963 | 0.25671 | 0.28902    | 0.24462 | 0.17968 | 0.29870                |
> |                  | +$\kappa$,$\alpha$      | 0.17084 | 0.25210 | 0.17031 | 0.29183    | 0.23460 | 0.19982 | 0.29493                |
> |                  | +$\phi$,$\psi$,$\omega$ | 0.21646 | 0.29070 | 0.21599 | 0.31065    | 0.23602 | 0.22435 | 0.31575                |

---

> > ### Author Response · Authors · 2024-11-20
> > **A kind rebuttal to Reviewer nr39 (Part 2)**
> >
> > **Q2:** "Also, the trend in this area of research has been towards developing large pre-trained models which can be fine-tuned for a range of tasks. I think this contribution would be strengthened by some exploration of the ability to scale the proposed method and how it compares to existing pre-trained structural encoders."
> >
> > **A2:** We agree with your perspective that developing large pre-trained models is highly valuable and represents a significant trend in this research area. Due to time and experimental constraints, our current work focuses on establishing a robustness protein representation learning framework. This is a foundational step towards addressing challenges related to noise or prediction errors in structural data.
> > In the future, we plan to extend $E^{3}$former to a pre-training framework, enabling it to leverage large-scale data and adapt to a broader range of tasks.
> >
> >
> >
> > **Q3:** Minor
> >
> > **A3:** We apologize for these errors. We have corrected these in the revised manuscript and all changes have been colored in blue.
> >
> >
> >
> > **Q4:** "How much hyperparameter tuning did the authors do? Particularly for the models that were not drawn from the ProteinWorkshop benchmark such as equiformer."
> >
> > **A4:** We did not conduct extensive hyperparameter tuning for the models. For components shared within the benchmark framework (e.g., learning rate, optimizer, dropout, and output head architecture), we applied consistent settings across all models to ensure a fair comparison. For model-specific configurations, such as Equiformer, we use most of the default settings provided in the original implementation and set the number of transformer blocks to 6. We hope this clarifies our approach to hyperparameter tuning.
> >
> >
> >
> > **Q5:** "L284: During each training or inference stage -> What does stage refer to here? Every epoch? Every batch? Or is the adjacency fixed for a given training run?"
> >
> > **A5:**  In this context, "stage" refers to **each batch** during the training or inference process. The adjacency is not fixed; instead, it is dynamically sampled and updated before each forward pass.
> >
> >
> >
> > **Q6:** "In Figure 4, a crystallographer would consider all of these resolutions extremely good -> good. I suggest expanding the range of resolutions considered"
> >
> > We agree that this could provide a broader perspective on the model's performance across different resolution ranges. However, the resolution distribution of the data sourced from PDB is uneven, with fewer samples available at low resolutions. To ensure that each resolution interval contains a sufficient number of samples for meaningful analysis, we decide to narrow the the range of resolutions. We also include a table showing the distribution of sample counts across different resolution in test dataset, we hope this can address your concern.
> >
> > |               | 0.0-1.5 | 1.5-2.0 | 2.0-2.5 | 2.5-3.0 | 3.0-3.5 | 3.5-9.0 |
> > | ------------- | ------- | ------- | ------- | ------- | ------- | ------- |
> > | CATH          | 102     | 383     | 298     | 114     | 34      | 0       |
> > | PPBS 70       | 23      | 178     | 171     | 113     | 43      | 17      |
> > | PPBS Homology | 105     | 428     | 508     | 284     | 129     | 21      |
> > | PPBS Topology | 77      | 327     | 313     | 157     | 29      | 8       |
> > | PPBS None     | 56      | 329     | 334     | 194     | 100     | 40      |
> > | PPBS All      | 261     | 1262    | 1326    | 748     | 301     | 86      |

---

> > > ### Author Response · Authors · 2024-11-20
> > > **A kind rebuttal to Reviewer nr39 (Part 3)**
> > >
> > > **Q7:**"Were the ablation studies hyperparameter tuned?"
> > >
> > > **A7:** In the ablation studies, we directly removed the corresponding modules without performing additional hyperparameter tuning.
> > >
> > >
> > >
> > > **Q8:**"What does the highlighting denote in Table 6?"
> > >
> > > **A8:** The highlighting in Table 6 denotes that the corresponding metric achieved the best performance among all compared methods. We have updated the manuscript to explicitly clarify this.
> > >
> > >
> > >
> > > **Q9:**"In the appendix, the authors mention there were difficulties mapping the MASIF site dataset labels to AF2 predictions. Could they please elaborate a little here?"
> > >
> > > **A9:** Thank you for raising this important question. We have elaborated on the challenges of mapping the MASIF dataset to AlphaFold-predicted structures.
> > >
> > > The primary challenge in mapping the MASIF dataset to AlphaFold-predicted structures arose during sequence alignment using the BLOSUM62 global alignment algorithm. We observed that the resulting alignments were highly fragmented, resulting in short, disconnected segments rather than continuous regions. Given these difficulties in obtaining reliable results, we made the decision to exclude this dataset from our benchmark experiments.
> > >
> > > Reference:
> > >
> > > [1] Kimanius, Dari, et al. "Data-driven regularization lowers the size barrier of cryo-EM structure determination." *Nature Methods* (2024): 1-6.
> > >
> > > [2] Chakravarty, Devlina, et al. "AlphaFold predictions of fold-switched conformations are driven by structure memorization." *Nature Communications* 15.1 (2024): 7296.
> > >
> > > [3]  Chakravarty, Devlina, and Lauren L. Porter. "AlphaFold2 fails to predict protein fold switching." *Protein Science* 31.6 (2022): e4353.

---

> > > > ### Comment · Reviewer_nr39 · 2024-11-27
> > > >
> > > > I thank the authors for their responses to my questions. I believe my score is appropriate and so I will maintain it. I think it is difficult to give strong evidence for all claims in the absence of proper hyperparameter tuning. Considering the results in tables two and three, for example, the absolute gaps in performance between the next best methods are not particularly large and --- to my mind --- certainly in the regime where additional hyperparameter tuning could alter the results.

---

> > > > > ### Author Response · Authors · 2024-12-02
> > > > > **A kind response to Reviewer nr39**
> > > > >
> > > > > Thank you for your response, and we understand your concern regarding the potential impact of hyperparameter tuning. We agree that more detailed tuning for each model could potentially yield better performance. Because of this, during the tuning process, we made an effort to maintain fairness by using consistent parameters wherever possible. For model-specific parameters and architectures, we tried to use standard configurations across all tasks to minimize discrepancies caused by tuning differences. Therefore, we believe that our model is capable of achieving consistent performance improvements on noisy datasets.
> > > > >
> > > > > Once again, we sincerely thank you for your feedback. We believe your insights will contribute significantly to improving the quality of this work.

---

### Official Review · Reviewer_A5FR · 2024-11-04

**Soundness:** 3
**Presentation:** 3
**Contribution:** 2
**Rating:** 6
**Confidence:** 4

**Summary:**

This work proposed an Energy-aware Elastic Equivariant Transformer-SSM hybrid architecture
for 3D macromolecular structure representation learning. The core of the model is to adaptively
utilize the energy-aware module to bulid proximity graph and use the equivariant SSM module to
express high-order features sparsely. The above improvements enable model more tolerant to data
deviation and noise.

**Strengths:**

1. The evaluation is complete and thorough. I appreciate the authors’ effort on using confidence split for validing the noise tolerence. A kind suggestion is that the split interval is a little narrow (80-85,90-95…), instead using 0-60, 60-70, 70-80…. may be more approporiate. And the readers may also want to know the performance under noisy structures (e.g. with confidence below 70, 60, etc)
2. The writing of introduction is good and clear

**Weaknesses:**

1. There are typos and incorrect spaces in the method section, in Eq,3,4, 10,11. I kindly suggest the authors to keep a clean format.
2. The connections between equivariant elastic selective SSM and rotation noise are week for me. It’s hard for me to get the logic from rotation noise to SSM. I kindly suggest the authors to introduce more details about this part.
3. Performance improvement on some tasks are a little marginal and the improvement introduced by the key component Energy-aware radius function is marginal. The motivation for using SSM is missing.

**Questions:**

1. can we achieve the noise tolerance via some simple data augumentation such as adding some noise to the original protein structure as done in ProteinMPNN?
2. What’s the running time of E^3former? Can E^3former be applied to protein property predition tasks?
3. Does the noise tolerence problem really exsit? Since the structure prediction accuracy will continue to improve and this tolerance should be easily maintained via some data augumentation and pretraining technique like Graph Structure Learning on proteins.  I kindly welcome the disccusions on this from the authors.

---

> ### Author Response · Authors · 2024-11-20
> **A kind rebuttal to Reviewer A5FR (Part 1)**
>
> **Q1:** "A kind suggestion is that the split interval is a little narrow (80-85,90-95…), instead using 0-60, 60-70, 70-80…. may be more appropriate. And the readers may also want to know the performance under noisy structures (e.g. with confidence below 70, 60, etc)"
>
> **A1:** Thank you for raising this valuable question. While using broader intervals (e.g., 0–60, 60–70, 70–80) seems reasonable, we decided to use narrower intervals (e.g., 80–85, 90–95) due to the specific data distribution. Our primary aim is to ensure that each interval contains a sufficient number of samples for meaningful statistical evaluation. The sample distribution across intervals is as follows:
>
> |                  | 00-80 | 80-85 | 85-90 | 90-95 | 95-100 |
> | ---------------- | ----- | ----- | ----- | ----- | ------ |
> | CATH-AF          | 157   | 118   | 239   | 350   | 151    |
> | PPBS-AF 70       | 56    | 47    | 70    | 115   | 190    |
> | PPBS-AF Homology | 133   | 120   | 169   | 415   | 541    |
> | PPBS-AF Topology | 77    | 39    | 72    | 189   | 475    |
> | PPBS-AF None     | 148   | 96    | 143   | 312   | 229    |
> | PPBS-AF All      | 414   | 302   | 454   | 1031  | 1435   |
>
> As seen, the number of samples in very low-confidence intervals is sparse. This makes it challenging to conduct reasonable evaluations.
>
> **Q2:** "There are typos and incorrect spaces in the method section, in Eq,3,4, 10,11. I kindly suggest the authors to keep a clean format."
>
> **A2:** Thank you for pointing out the typographical errors. The corresponding sections have been carefully revised, and all changes are visible in the revised manuscript.
>
> **Q3:** "The connections between equivariant elastic selective SSM and rotation noise are week for me. It’s hard for me to get the logic from rotation noise to SSM. I kindly suggest the authors to introduce more details about this part."
>
> **A3:** Thank you for highlighting the need for a clearer explanation of the connection between equivariant elastic selective SSM and rotation noise. Below, we provide a more detailed explanation.
>
> **Equivariant Models and Susceptibility to Rotation Noise:**
> Equivariant models, particularly those handling high-order tensors, are more sensitive to rotation noise [1,2]. This is because rotation noise introduces distortions in the geometric relationships captured by these high-dimensional tensors, significantly impacting the model's performance. As a result, addressing rotation noise is critical for maintaining the accuracy of equivariant architectures.
>
> **Role of Elastic Selective SSM:**
> The elastic selective SSM serves as a sparse alternative to traditional Transformer attention mechanisms [3]. In our framework, it acts as a filter for high-dimensional tensor signals, selectively focusing on stable, informative components while omitting noisy or less relevant information. By increasing the signal-to-noise ratio, the elastic selective SSM effectively mitigates the impact of rotation noise on the model's performance.
>
> **Q4:** "Performance improvement on some tasks are a little marginal and the improvement introduced by the key component Energy-aware radius function is marginal. The motivation for using SSM is missing."
>
> **A4:** The performance gains introduced by our model are more pronounced in situation with significant noise. In **low-Confidence AlphaFold-predicted samples,** as shown in Figure 3 and Table 6, the model demonstrates notable improvements, with performance increases exceeding 20% in the PPBS-AF Homology dataset for samples with 0–80% confidence. While the performance gains are relatively smaller in experimental datasets, the model consistently maintains state-of-the-art performance, demonstrating its robustness across diverse data conditions.
> The motivation for using the SSM module lies in its ability to process high-order tensors with a higher signal-to-noise ratio. By selectively focusing on critical geometric features, the SSM module reduces the influence of noise and enables more effective representation learning. This has been further detailed in our response to **[Q3]**.

---

> > ### Author Response · Authors · 2024-11-20
> > **A kind rebuttal to Reviewer A5FR (Part 2)**
> >
> > **Q5:** "can we achieve the noise tolerance via some simple data augmentation such as adding some noise to the original protein structure as done in ProteinMPNN? "
> >
> > **A5:** Thank you for your thoughtful question. The answer is that it cannot be fully achieved. While simple data augmentation methods, such as adding noise to the original protein structure as done in ProteinMPNN[7], may work to some extent for model-predicted structures, their effectiveness for experimental data remains uncertain. This is because experimental data often contains implicit structural information, and augmentation could potentially disrupt such features. We have provided a more extensive discussion on this topic, including relevant experiments and observations, in our response to **[Q7]**.
> >
> > Compare to the method depending heavily on the distribution of the artificially added noise, $E^{3}$former achieves noise tolerance without relying on inductive biases tailored to specific types of noise. Instead, it employs the elastic selective mechanism focuses on filtering out noisy signals and leverages molecular dynamics-inspired Energy-aware radius graphs to correct the noise by utilizing the interdependence of macromolecules. Both of them cannot be directly replaced by data augmentation alone.
> >
> > **Q6:** "What’s the running time of E^3former? Can E^3former be applied to protein property prediction tasks?"
> >
> > The running time of the mentioned model test in CATH dataset with one A100(80GB) are list as follows:
> >
> > |           | EGNN | TFN  | Schnet | GearNet | GCPNet | Equiformer | E3former | Sample nums | Batchsize |
> > | --------- | ---- | ---- | ------ | ------- | ------ | ---------- | -------- | ----------- | --------- |
> > | Training  | 115s | 712s | 47s    | 141s    | 239s   | 328s       | 387s     | 18007       | 32        |
> > | Inference | 13s  | 17s  | 9s     | 16s     | 25s    | 21s        | 28s      | 931         | 32        |
> >
> > Despite being slower due to the incorporation of equivariant neural network components and the Transformer architecture, $E^{3}$former is comparable to other Transformer-based methods,while maintaining a runtime similar to existing transform-style approaches.
> >
> > As a general-purpose protein encoder,  $E^{3}$former is also applicable to graph-level tasks such as protein property prediction. Our current work mainly focuses on evaluating its performance on node-level tasks, which are critical for understanding fine-grained structural details. In the future, we plan to extend $E^{3}$former to a broader range of tasks, including protein property prediction, to further explore its potential applications.

---

> > > ### Author Response · Authors · 2024-11-20
> > > **A kind rebuttal to Reviewer A5FR (Part 3)**
> > >
> > > **Q7:** "Does the noise tolerance problem really exist? Since the structure prediction accuracy will continue to improve and this tolerance should be easily maintained via some data augmentation and pretraining technique like Graph Structure Learning on proteins. I kindly welcome the discussion on this from the authors."
> > >
> > > **A7:** Thank you for this constructive question. We have given this topic considerable thought and are delight to share our perspective on the noise tolerance problem and its relation to pretraining and data augmentation.
> > >
> > > **1. Pretraining for Noise Tolerance**
> > > We agree that pretraining can significantly enhance model performance and noise tolerance. For example, in 2022, Zhang et al. pre-trained their models on AlphaFold-predicted structures[8], and the resulting models demonstrated excellent generalization performance across various protein property prediction tasks. This aligns with your observation that leveraging large-scale, high-quality protein structure data for pretraining can endow models with noise tolerance. However, pretraining is a double-edged sword, as it requires enormous computational resources and access to a large amount of high-quality protein structure data, which represents a significant cost. The benefits of achieving noise resistance through pretraining must be carefully weighed against these substantial resource demands, making it a trade-off that requires thoughtful consideration.
> > >
> > > **2. Data Augmentation Limitations**
> > > While data augmentation can mitigate noise in some cases, its effectiveness heavily depends on the task and the type of data. In some instances, excessive augmentation can degrade model performance. This is supported by findings from ProteinMPNN[8] in the context of protein inverse folding. Dauparas et al. applied Gaussian noise (variance = 0.02Å) to backbone atoms during training for both experimentally determined structures and AlphaFold2-predicted structures. Their results showed that while performance improved significantly with augmentation for AlphaFold2-predicted structures, it worsened for experimentally determined structures, suggesting that augmentation can disrupt hidden structural features present in high-precision data.
> > >
> > > The authors pointed that experimentally resolved structures inherently encode indirect information about amino acid types through backbone coordinates (e.g., specific angles or distances common to certain amino acids in specific environments). This implicit information can be exploited by models during training, improving performance. However, such implicit features are absent in predicted structures, such as those from AlphaFold2, where side-chain information is often incomplete, and backbone flexibility is limited. Therefore, applying data augmentation to high-resolution experimental structures may degrade model performance by disrupting these hidden information, whereas it benefits predicted structures by introducing variability.
> > >
> > > **3. Supporting Experiments on Data Augmentation**
> > > To validate the above observations, we conducted additional experiments using the fastest baselines, **EGNN** and **SchNet**, as well as **Equiformer**, which shares a similar Transformer architecture with $E^3$former. We performed these experiments on the PPBS dataset. Following the methodology in ProteinMPNN[8], we added Gaussian noise (variance = 0.02Å) to the backbone atoms during training.
> > >
> > > The results, as shown in the table, indicate that training with data augmentation led to a decline in performance for all models compared to the original results presented in our paper. This aligns with the analysis above, further confirming that data augmentation can negatively affect models when applied to high-resolution experimental data.
> > > |               | Features | EGNN    | Schnet  | Equiformer | $E^3\text{former}$ |
> > > | ------------- | -------- | ------- | ------- | ---------- | ------------------ |
> > > | PPBS 70       | +Seq     | 0.64825 | 0.52087 | 0.66842    | 0.67213            |
> > > | PPBS Homology | +Seq     | 0.62216 | 0.47281 | 0.63977    | 0.62532            |
> > > | PPBS Topology | +Seq     | 0.65151 | 0.46895 | 0.69923    | 0.71092            |
> > > | PPBS None     | +Seq     | 0.53146 | 0.39184 | 0.53041    | 0.56724            |
> > > | PPBS All      | +Seq     | 0.62477 | 0.44114 | 0.64853    | 0.66721            |

---

> > > > ### Author Response · Authors · 2024-11-20
> > > > **A kind rebuttal to Reviewer A5FR (Part 4)**
> > > >
> > > > **Future Directions:**
> > > > While our current experiments are relatively preliminary, we are actively extending this work to include datasets like PPBS-AF and exploring the effects of data augmentation on protein data from different sources. We hope this will lead to a deeper understanding of the relationship between data augmentation, noise, and structural properties. In future work, we aim to provide more robust experimental and theoretical insights to further solidify these findings.
> > > >
> > > > Thank you for raising this question, which has encouraged us to refine and expand our analysis of noise tolerance in protein structure learning. We hope our response addresses your concerns and provides useful context for this discussion.
> > > >
> > > > Reference:
> > > >
> > > > [1] Li, Zian, et al. "Is distance matrix enough for geometric deep learning?." *Advances in Neural Information Processing Systems* 36 (2024).
> > > >
> > > > [2] Duval, Alexandre, et al. "A Hitchhiker's Guide to Geometric GNNs for 3D Atomic Systems." *arXiv preprint arXiv:2312.07511* (2023).
> > > >
> > > > [3] Behrouz, Ali, and Farnoosh Hashemi. "Graph mamba: Towards learning on graphs with state space models." *Proceedings of the 30th ACM SIGKDD Conference on Knowledge Discovery and Data Mining*. 2024.
> > > >
> > > > [4] Chakravarty, Devlina, et al. "AlphaFold predictions of fold-switched conformations are driven by structure memorization." *Nature Communications* 15.1 (2024): 7296.
> > > >
> > > > [5] Chakravarty, Devlina, and Lauren L. Porter. "AlphaFold2 fails to predict protein fold switching." *Protein Science* 31.6 (2022): e4353.
> > > >
> > > > [6] Beton, Joseph G., et al. "Cryo-EM structure and B-factor refinement with ensemble representation." *Nature Communications* 15.1 (2024): 444.
> > > >
> > > > [7] Dauparas, Justas, et al. "Robust deep learning–based protein sequence design using ProteinMPNN." *Science* 378.6615 (2022): 49-56.
> > > >
> > > > [8] Zhang, Zuobai, et al. "Protein representation learning by geometric structure pretraining." arXiv preprint arXiv:2203.06125 (2022).

---

> > > > > ### Author Response · Authors · 2024-12-02
> > > > > **A Kind Reminder to Reviewer A5FR**
> > > > >
> > > > > Thank you for your feedback. We have made significant efforts to explore the Noise Tolerance issues you mentioned and look forward to receiving your further feedback.

---

> > > > > > ### Comment · Reviewer_A5FR · 2024-12-03
> > > > > >
> > > > > > Sorry for the late reply. After reading the revised manuscript and the comments from other reviewers, I agree with Reviewer GWEV that the issue of noise tolerance should be presented better. As for the pre-training methods for noise tolerance, I think there are other pre-training methods specifically designed to improve robustness (e.g. Graph Structure Learning). Comparison with these methods is not required, as E3former lies in the backbone design field. I just wish the authors could make a more complete survey about the noise tolerance issue of predicted structure and methods to address it. I appreciate the authors' rebuttal very much, and I raise my score to 6. Sorry again for the late reply.

---

> > > > > > > ### Author Response · Authors · 2024-12-03
> > > > > > > **A kind response to Reviewer A5FR**
> > > > > > >
> > > > > > > I sincerely thank you for your thoughtful response! Your suggestions are valuable in improving the quality of our work.
> > > > > > >
> > > > > > > Here, I would like to present a survey on the noise tolerance issues in predicted protein structures and the methods developed to address these challenges.
> > > > > > >
> > > > > > > ### **Noise in Predicted Structures**
> > > > > > >
> > > > > > > Various works on noise tolerance in predicted structures address different types of noise, which can be classified as follows:
> > > > > > >
> > > > > > > 1. **Perturbation Noise:**
> > > > > > >    This type of noise is typically approximated as Gaussian-distributed fluctuations, primarily caused by measurement inaccuracies of experimental instruments, molecular dynamics, or environmental factors [6].
> > > > > > > 2. **Flexible Regions Noise:**
> > > > > > >    AlphaFold and other structure prediction methods often struggle to accurately capture the structural variations of flexible regions such as loops and disordered regions [1].
> > > > > > > 3. **Long-Range Interaction Noise:**
> > > > > > >    AlphaFold may exhibit spatial deviations, especially when predicting long-range interactions in multi-domain proteins or large molecular complexes, leading to some degree of structural bias [1].
> > > > > > > 4. **Template Noise:**
> > > > > > >    When using template-based methods, structural deviations or incomplete templates can result in inaccurate predictions. Some methods enhance data to increase robustness against template bias, particularly when dealing with low-resolution or incomplete templates [7].
> > > > > > > 5. **AlphaFold Structures Side-Chain Information Loss:**
> > > > > > >    AlphaFold primarily preserves backbone information, but loses fine-grained details about side-chain information [2].
> > > > > > >
> > > > > > > ### **Methods to Address Noise**
> > > > > > >
> > > > > > > A reliable approach to addressing noise is to use **Graph Structure Learning** to capture spatial constraints in protein structures, which improves prediction accuracy, particularly when dealing with high flexibility, complex topologies, or low-resolution input data [1]. Additionally, methods have been developed to mitigate noise by:
> > > > > > >
> > > > > > > - **Data Augmentation** by adding noise to AlphaFold-predicted structures [2],
> > > > > > > - **Refinement using multiple large models** to further optimize structural predictions [3],
> > > > > > > - **Domain-specific finetune** to address specific types of noise [4],
> > > > > > > - **Using denoising networks** to reduce noise and prevent the prediction of physically implausible structures through adversarial loss  [5].
> > > > > > >
> > > > > > > Here is the Table of Relevant Work on Noise Tolerance in Predicted Structures:
> > > > > > >
> > > > > > > | **Noise type/Method**              | **Graph Structure Learning [1]** | **Data Augmentation [2]** | **Data Refinement [3]** | **Pretrain or Finetune [4]** | **Denoising Network [5]** |
> > > > > > > | ---------------------------------- | ------------------------------- | ------------------------ | ---------------------- | --------------------------- | ------------------------- |
> > > > > > > | **Perturbation Noise**             | Related                         | Related                  | Weak Related           | Weak Related                | Related                   |
> > > > > > > | **Sideschain Information Missing** | Weak Related                    | Related                  | Weak Related           | Related                     | Weak Related              |
> > > > > > > | **Flexible Regions Noise**         | Related                         | Weak Related             | Related                | Weak Related                | Related                   |
> > > > > > > | **Template Noise**                 | Related                         | Weak Related             | Related                | Related                     | Weak Related              |
> > > > > > > | **Long-Range Interaction Noise**   | Related                         | Weak Related             | Related                | Related                     | Weak Related              |
> > > > > > >
> > > > > > > Thank you once again for your insightful feedback, which has greatly deepened our understanding of this topic.
> > > > > > >
> > > > > > > **Reference**
> > > > > > >
> > > > > > > [1] Huang, Yufei, et al. "Protein 3d graph structure learning for robust structure-based protein property prediction." *Proceedings of the AAAI Conference on Artificial Intelligence*. Vol. 38. No. 11. 2024.
> > > > > > >
> > > > > > > [2] Dauparas, Justas, et al. "Robust deep learning–based protein sequence design using ProteinMPNN." *Science* 378.6615 (2022): 49-56.
> > > > > > >
> > > > > > > [3] Frank, Christopher Josef, et al. "Alphafold2 refinement improves designability of large de novo proteins." *bioRxiv* (2024): 2024-11.
> > > > > > >
> > > > > > > [4] Oda, Toshiyuki. "Refinement of AlphaFold-Multimer structures with single sequence input." *bioRxiv* (2022): 2022-12.
> > > > > > >
> > > > > > > [5] Liu, Yufeng, et al. "De novo protein design with a denoising diffusion network independent of pretrained structure prediction models." *Nature Methods* (2024): 1-10.
> > > > > > >
> > > > > > > [6] Pöschko, Maria Theresia, et al. "Nonlinear detection of secondary isotopic chemical shifts in NMR through spin noise." *Nature communications* 8.1 (2017): 13914.
> > > > > > >
> > > > > > > [7] Huang, Bin, et al. "Protein structure prediction: challenges, advances, and the shift of research paradigms." *Genomics, Proteomics & Bioinformatics* 21.5 (2023): 913-925.

---

### Official Review · Reviewer_zNqp · 2024-11-04

**Soundness:** 2
**Presentation:** 2
**Contribution:** 2
**Rating:** 6
**Confidence:** 4

**Summary:**

The paper presents $E^3$former, a novel protein representation model that combines an equivariant Transformer and State Space Model (SSM) architecture. $E^3$former aims at enhancing robustness against noise in protein structure data, it employs an energy-aware radius graph to dynamically adapt to node proximity and a high-tensor elastic selective SSM to efficiently capture geometric features. Experimental results indicate that $E^3$former surpasses existing models in tasks like protein binding site prediction and inverse folding, especially when dealing with AlphaFold-predicted noisy data.

**Strengths:**

- The paper proposes an innovative hybrid Transformer-SSM model that addresses noise issues in protein structure data.
- It implements an energy-aware radius function to adaptively construct protein graphs based on node proximity to mitigate data biases.
- It demonstrates substantial performance improvements over SOTA models across multiple datasets and tasks.

**Weaknesses:**

- The captions of the figures and tables are too short (i.e., fig. 2), it is too difficult to understand and follow. The terminologies and notations are also not clarified, which is very confusing.

- While the model demonstrates efficacy on AlphaFold and crystal structures, this reliance could constrain its generalizability to other types of protein data with different structural features.

- In  Section 4.2, the rationale for selecting specific baseline models and configurations lacks clarity. The reasons of such implementation should be explained and introduced with sufficient motivation.

- The results analysis and discussions are insufficient, like comparison across baseline models. For example, in Tables 2 and 3, the authors should discuss why the baseline results differ from the proposed model and what factors may contribute to these discrepancies. In addition, in Table 3, the addition of certain features (e.g., ($\kappa$), ($\alpha$)) resulted in decreased performance in some tasks, yet the reasons for this are not explored.

**Questions:**

- In the energy-aware radius graph module, the authors chose fixed parameters (($\epsilon$) and ($\sigma$)). Can the authors provide more detail on the rationale for these choices? What impact do these parameters have on results across different datasets? Has experimentation been conducted with alternative parameter combinations?
- In Table 3, the addition of certain features (e.g., ($\kappa$), ($\alpha$)) led to performance declines in some tasks. Can the authors elaborate on this phenomenon?
- In Figure 3, the authors state: "As illustrated in Figure 3, our model demonstrates a more substantial performance enhancement when the confidence is reduced." However, this does not seem convincing based on the figure. For instance, while $E^3$former’s AUPRC slightly outperforms the baseline in the 90-100 confidence interval, it is lower than the baseline in the 85-90 interval. More comprehensive results should be included, like reporting box plots for each confidence interval would be more informative.
- Based on Table 2 and Figure 3 (PPBS-AF All dataset AUPRC), there may be a long-tail problem in the results. Can the authors present the outcomes for the top 10 examples within the PPBS-AF All dataset to explore this further?

---

> ### Author Response · Authors · 2024-11-20
> **A kind rebuttal to Reviewer zNqp (Part 1)**
>
> We greatly appreciate your constructive comments and the valuable questions you raised. We are also grateful for your recognition of the innovation and value of our two core modules: the hybrid Transformer-SSM and the Energy-aware radius function.
>
> **Q1**: "The captions of the figures and tables are too short (i.e., fig. 2), it is too difficult to understand and follow. The terminologies and notations are also not clarified, which is very confusing."
>
> **A1**: We have revised and clarified the captions in the manuscript. All updated captions are colored in blue.
>
> **Q2:** "While the model demonstrates efficacy on AlphaFold and crystal structures, this reliance could constrain its generalizability to other types of protein data with different structural features."
>
> **A2**: To clarify, $E^{3}$former can generalize to other types of structural data beyond the datasets explored in this study. In our work, we focus on two representative types of structural data: (1) experimentally measured data, such as crystal structures, and (2) model-generated data, such as AlphaFold-predicted structures. However, the inductive biases of our model are not limited to these two data types. From molecular dynamics perspective, the positional information of atoms within macromolecules is interdependent, providing latent unsupervised signals for self-correction(Energy-aware graph). Noise in structural data often propagate and intensify within equivariant neural networks. By selectively leveraging these signals, we effectively reduce the impact of such noise(Equivariant high-tensor-elastic SSM). These assumptions are grounded in the properties of biological macromolecules rather than limited to any specific data type.
>
> **Q3:** "In Section 4.2, the rationale for selecting specific baseline models and configurations lacks clarity. The reasons of such implementation should be explained and introduced with sufficient motivation."
>
> **A3:**  We provide additional details and rationale for our choices as follow.
> Our choices based on a recent benchmark for evaluating models in protein representation learning [1]. To ensure fairness in our comparisons, we use the default settings from this work for most baselines and common configurations, such as learning rates, optimizers, and architectural parameters.
> Additionally, we selected baseline models based on their representativeness and characteristics. The chosen baselines include a mix of classical and state-of-the-art approaches: **SchNet** (Classical Invariant GNNs) [3], **TFN** (Classical Spherical Equivariant GNNs) [4], **EGNN** (Equivariant GNNs) [5], **Equiformer** (Spherical Equivariant GNNs with Transformer blocks) [6], **GearNet** (Powerful protein-specific SE(3) Invariant GNNs) [7], **GCPNet** (Powerful protein-specific Equivariant GNNs) [8]. We hope this additional information helps clarify our choices.
>
> **Q4:** "The results analysis and discussions are insufficient, like comparison across baseline models. For example, in Tables 2 and 3, the authors should discuss why the baseline results differ from the proposed model and what factors may contribute to these discrepancies.
>
> **A4:** Our model demonstrates superior performance compared to baselines due to two key factors:
>
> 1. **Stronger Fitting Capability with the Hybrid Transformer-SSM Architecture:**
>    The hybrid architecture enhances the model’s fitting ability, enabling it to achieve more significant performance improvements in challenging tasks. When fewer features are used (+Seq features) or in situation with large training-test set difference, such as the "none" split in the PPBS dataset, the model’s advantages are more pronounced (Tables 2 and 3).
>
> 2. **Noise Resilience through Selective SSM and Adaptive Energy Functions:**
>    By incorporating Selective SSM and an Adaptive Energy function, the model effectively enhances its robustness to noise. In noisy datasets, such as the AlphaFold-predicted data (Table 2) and in low-resolution interval with significant noise effects (Figure 4), these components allow the model to demonstrate more comparable performance.

---

> > ### Author Response · Authors · 2024-11-20
> > **A kind rebuttal to Reviewer zNqp (Part 2)**
> >
> > **Q5**: "In addition, in Table 3, the addition of certain features (e.g., (κ), (α)) resulted in decreased performance in some tasks, yet the reasons for this are not explored."
> >
> > **A5**: This is a good question, and one that we have also considered in detail. In Lines 452–455, we hypothesize that the decreased performance may be caused by **information redundancy** and **noise amplification**. This phenomenon has also been observed in another protein benchmark study [1], indicating that it is not unique to our model.
> > To better understand this, we analyzed the features mentioned in question, which were computed using the `graphein.protein.tensor.angles` library:
> >
> > - **Alpha Angle:** Calculated by measuring the angle between vectors formed by three consecutive $C_{\alpha}$ atoms in a protein's backbone.
> > - **Kappa Angle:** Determined by computing the dihedral angle defined by four consecutive $C_{\alpha}$ atoms, reflecting the backbone’s torsion.
> >
> > While these features provide valuable geometric insights, they also may overlap with existing structural features (e.g., 3D coordinates), leading to redundant information. Besides, in noisy datasets, such as AlphaFold predictions, these features can act as shortcuts, making the model easy to overfit.
> >
> > **Q6**: "In the energy-aware radius graph module, the authors chose fixed parameters ((ϵ) and (σ)). Can the authors provide more detail on the rationale for these choices? What impact do these parameters have on results across different datasets? Has experimentation been conducted with alternative parameter combinations?"
> >
> > **A6:** We chose fixed parameters ($\epsilon$ and $\sigma$) for all tasks based on their performance during early-stage experiments. This decision was motivated by our observation that the model is insensitive to these parameters within a reasonable range. We fixed parameters across all datasets to highlight the model's robustness to hyperparameter selection.
> > To further validate this, we conduct a parameter sensitivity experiment on the CATH dataset. The results below indicate that varying $\epsilon$ and $\sigma$ within reasonable ranges has little impact on the model's performance, demonstrating its robustness to these parameters.
> >
> > |              | $ \epsilon=3$ | $\epsilon=3.8$ | $\epsilon=5$ |
> > | ------------ | ------------- | -------------- | ------------ |
> > | $\sigma=0.5$ | 7.57          | 7.54           | 7.61         |
> > | $\sigma=1$   | 7.51          | 7.53           | 7.53         |
> > | $\sigma=1.5$ | 7.58          | 7.56           | 7.57         |
> >
> > **Q7**: "In Table 3, the addition of certain features (e.g., (κ), (α)) led to performance declines in some tasks. Can the authors elaborate on this phenomenon?"
> >
> > **A7:** Thank you for highlighting this question. We have addressed this phenomenon in detail in our response to **[Q5]**, where we discussed how the addition of certain features can lead to performance declines in some tasks. Specifically, this behavior is attributed to potential information redundancy with existing features (e.g., 3D coordinates) and noise amplification in noisy datasets, which may cause the model to overfit.

---

> > > ### Author Response · Authors · 2024-11-20
> > > **A kind rebuttal to Reviewer zNqp (Part 3)**
> > >
> > > **Q8**: "In Figure 3, the authors state: "As illustrated in Figure 3, our model demonstrates a more substantial performance enhancement when the confidence is reduced." However, this does not seem convincing based on the figure. For instance, while E3former’s AUPRC slightly outperforms the baseline in the 90-100 confidence interval, it is lower than the baseline in the 85-90 interval. More comprehensive results should be included, like reporting box plots for each confidence interval would be more informative."
> > >
> > > **A8:** Thank you for your detailed feedback regarding Figure 3 and the need for more comprehensive results . Below, we will address your concerns and provide additional context.
> > >
> > > **Performance in the 85–90 Confidence Interval:**
> > > The performance decline of our model in the 85–90 confidence interval is related to the data distribution within this range. As shown in the table below, we analyzed the number and type of test samples across different confidence intervals. In the 85–90 interval, the total number of samples is relatively small, and a significant proportion of these samples belong to the PPBS-70 dataset. The randomness introduced in this subset impacts the performance gain demonstrated by our model in this interval.
> > >
> > > **Comprehensive Results:**
> > > Since our model exhibits low prediction in Tables 2 and 3, instead of box plot, we have taken the following steps to provide a more comprehensive results. We have conducted experiments comparing model performance across different features, tasks, and confidence intervals (Appendix C.1, Table 6). Besides, for Figure 3, we will supplement these results with the specific sample numbers here to provide a more comprehensive results.
> > > |                  | 00-80 | 80-85 | 85-90 | 90-95 | 95-100 |
> > > | ---------------- | ----- | ----- | ----- | ----- | ------ |
> > > | CATH-AF          | 157   | 118   | 239   | 350   | 151    |
> > > | PPBS-AF 70       | 56    | 47    | 70    | 115   | 190    |
> > > | PPBS-AF Homology | 133   | 120   | 169   | 415   | 541    |
> > > | PPBS-AF Topology | 77    | 39    | 72    | 189   | 475    |
> > > | PPBS-AF None     | 148   | 96    | 143   | 312   | 229    |
> > > | PPBS-AF All      | 414   | 302   | 454   | 1031  | 1435   |
> > >
> > > **Q9:** "Based on Table 2 and Figure 3 (PPBS-AF All dataset AUPRC), there may be a long-tail problem in the results. Can the authors present the outcomes for the top 10 examples within the PPBS-AF All dataset to explore this further?"
> > >
> > > **A9:** From the sample distribution outlined in **[A8]**, it is clear that AlphaFold-predicted structures exhibit a higher proportion of high-confidence predictions. Directly dividing the data uniformly by confidence intervals (e.g., 0–20, ..., 80–100) could lead to significant long-tail effects. This is why we choose narrower distribution intervals (e.g., 0–80, 80–85, 85–90, 90–95, 95–100) to mitigate this issue as much as possible.
> > >
> > > Regarding your question on the "top 10 examples," we are not certain if we fully understood your intent. Based on our understanding, we provide predictions for 10 representative cases in this anonymous url: **https://drive.google.com/file/d/1yu1nWNcS7zP4X2jmO8OblAF0l3FdCOSG/view?pli=1**. Please let us know if this match your expectations or if further clarification is required.
> > >
> > > [1] Jamasb, Arian R., et al. "Evaluating representation learning on the protein structure universe." *ArXiv* (2024).
> > >
> > > [2] Duval, Alexandre, et al. "A Hitchhiker's Guide to Geometric GNNs for 3D Atomic Systems." *arXiv preprint arXiv:2312.07511* (2023).
> > >
> > > [3] Schütt, Kristof T., et al. "Schnet–a deep learning architecture for molecules and materials." *The Journal of Chemical Physics* 148.24 (2018).
> > >
> > > [4] Thomas, Nathaniel, et al. "Tensor field networks: Rotation-and translation-equivariant neural networks for 3d point clouds." *arXiv preprint arXiv:1802.08219* (2018).
> > >
> > > [5] Satorras, Vıctor Garcia, Emiel Hoogeboom, and Max Welling. "E (n) equivariant graph neural networks." *International conference on machine learning*. PMLR, 2021.
> > >
> > > [6] Liao, Yi-Lun, and Tess Smidt. "Equiformer: Equivariant graph attention transformer for 3d atomistic graphs." *arXiv preprint arXiv:2206.11990* (2022).
> > >
> > > [7] Zhang, Zuobai, et al. "Protein representation learning by geometric structure pretraining." *arXiv preprint arXiv:2203.06125* (2022).
> > >
> > > [8] Morehead, Alex, and Jianlin Cheng. "Geometry-complete perceptron networks for 3d molecular graphs." *Bioinformatics* 40.2 (2024): btae087.
> > >
> > > [9] Schauperl, Michael, et al. "Data-driven analysis of the number of Lennard–Jones types needed in a force field." *Communications chemistry* 3.1 (2020): 173.

---

### Author Response · Authors · 2024-11-22
**General Response to Reviewers**

We sincerely thank all reviewers for your thorough and insightful feedback. We have provided detailed responses to each of your concerns and conducted additional experiments to address them. We hope this approach helps to alleviate your concerns. We are looking forward to your further feedback and will gladly address any unclear points in our responses.

---

### Author Response · Authors · 2024-11-25
**A Heartfelt Thanks to Our Reviewers**

We thank all reviewers for the insightful comments on our paper. Based on your feedback, we have further reflected on the model's noise tolerance and related challenges. Additionally, we conducted additional experiments, including cross-dataset evaluations, where we were surprised to find that the motivation and effectiveness of our model were further demonstrated in the process.

We have carefully addressed each reviewer's concerns in our responses. We sincerely appreciate your feedback and we are willing to clarify further or explore ways to improve our work.

---

### Meta-Review · Area_Chair_3Z9R · 2024-12-18

**Metareview:**

The paper introduces E^3former, a hybrid Transformer-SSM model for protein representation learning that aims to enhance robustness against noise in protein structure data. It uses an energy-aware radius graph and high-tensor elastic selective SSM to capture geometric features and adapt to complex atom interactions.

Strengths include innovative model architecture, empirical performance improvements over SOTA models, and a novel approach to handling noise in protein structures.

Weaknesses include lack of a thorough analysis of noise, the absence of proper hyperparameter tuning, and potential limitations in generalizability.

The decision to reject is based on the lack of thorough analysis of noise in protein structures, unclear motivation for the model, and the absence of proper hyperparameter tuning, which raises concerns about the validity of the conclusions.

**Additional Comments On Reviewer Discussion:**

Reviewer nr39 raised the problem that additional hyperparameter tuning could alter the results. Reviewer GWEV thinks that the lack of a thorough analysis of this noise raises significant concerns about the validity of the conclusions.

---

### Decision · Program_Chairs · 2025-01-22

Reject